# FINETUNING TEXT-TO-IMAGE DIFFUSION MODELS FOR FAIRNESS

**Xudong Shen**[*1], **Chao Du**[†2], **Tianyu Pang**[†2], **Min Lin**[2]
**Yongkang Wong**[3], **Mohan Kankanhalli**[†3]
[1]ISEP programme, NUS Graduate School, National University of Singapore
[2]Sea AI Lab, Singapore
[3]School of Computing, National University of Singapore
`xudong.shen@u.nus.edu; {duchao, tianyupang, linmin}@sea.com;`
`yongkang.wong@nus.edu.sg; mohan@comp.nus.edu.sg`

## ABSTRACT

The rapid adoption of text-to-image diffusion models in society underscores an urgent need to address their biases. Without interventions, these biases could propagate a skewed worldview and restrict opportunities for minority groups. In this work, we frame fairness as a distributional alignment problem. Our solution consists of two main technical contributions: (1) a distributional alignment loss that steers specific characteristics of the generated images towards a user-defined target distribution, and (2) adjusted direct finetuning of diffusion model's sampling process (adjusted DFT), which leverages an adjusted gradient to directly optimize losses defined on the generated images. Empirically, our method markedly reduces gender, racial, and their intersectional biases for occupational prompts. Gender bias is significantly reduced even when finetuning just five soft tokens. Crucially, our method supports diverse perspectives of fairness beyond absolute equality, which is demonstrated by controlling age to a $75\%$ young and $25\%$ old distribution while simultaneously debiasing gender and race. Finally, our method is scalable: it can debias multiple concepts at once by simply including these prompts in the finetuning data. We share code and various fair diffusion model adaptors at https://sail-sg.github.io/finetune-fair-diffusion/.

## 1 INTRODUCTION

Text-to-image (T2I) diffusion models (Nichol et al., 2021; Saharia et al., 2022) have witnessed an accelerated adoption by corporations and individuals alike. The scale of images generated by these models is staggering. To provide a perspective, DALL-E 2 (Ramesh et al., 2022) is used by over one million users (Bastian, 2022), while the open-access Stable Diffusion (SD) (Rombach et al., 2022) is utilized by over ten million users (Fatunde & Tse, 2022). These figures will continue to rise.

However, this influx of content from diffusion models into society underscores an urgent need to address their biases. Recent scholarship has demonstrated the existence of occupational biases (Seshadri et al., 2023), a concentrated spectrum of skin tones (Cho et al., 2023), and stereotypical associations (Schramowski et al., 2023) within diffusion models. While existing diffusion debiasing methods (Friedrich et al., 2023; Bansal et al., 2022; Chuang et al., 2023; Orgad et al., 2023) offer some advantages, such as being lightweight, they struggle to adapt to a wide range of prompts. Furthermore, they only approximately remove the biased associations but do not offer a way to *control* the distribution of generated images. This is concerning because perceptions of fairness can vary across specific issues and contexts; absolute equality might not always be the ideal outcome.

We frame fairness as a distributional alignment problem, where the objective is to align particular attributes of the generated images, such as gender, with a user-defined target distribution. Our solution consists of two main technical contributions. First, we design a loss function that steers the generated images towards the desired distribution while preserving image semantics. A key component is the distributional alignment loss (DAL). For a batch of generated images, DAL uses pre-trained classifiers to estimate class probabilities (*e.g.*, male and female probabilities) and dynamically gen-

---

[*]Work done during internship at Sea AI Lab. [†]Corresponding authors.

erates target classes that match the target distribution and have the minimum transport distance. To preserve image semantics, we regularize CLIP (Radford et al., 2021) and DINO (Oquab et al., 2023) similarities between images generated by the original and finetuned models.

Second, we propose adjusted direct finetuning of diffusion models, adjusted DFT for short, illustrated in Fig. 2. While most diffusion finetuning methods (Gal et al., 2023; Zhang & Agrawala, 2023; Brooks et al., 2023; Dai et al., 2023) use the same denoising diffusion loss from pre-training, DFT aims to directly finetune the diffusion model's sampling process to minimize any loss defined on the generated images, such as ours. However, we show the exact gradient of the sampling process has exploding norm and variance, rendering the naive DFT ineffective (illustrated in Fig. 1). Adjusted DFT leverages an adjusted gradient to overcome these issues. It opens venues for more refined and targeted diffusion model finetuning and can be applied for objectives beyond fairness.

Empirically, we show our method markedly reduces gender, racial, and their intersectional biases for occupational prompts. The debiasing is effective even for prompts with unseen styles and contexts, such as "*A philosopher reading. Oil painting*" and "*bartender at willard intercontinental makes mint julep*" (Fig. 3). Our method is adaptable to any component of the diffusion model being finetuned. Ablation study shows that finetuning the text encoder while keeping the U-Net unchanged hits a sweet spot that effectively mitigates biases and lessens potential negative effects on image quality. Surprisingly, finetuning as few as five soft tokens as a prompt prefix is able to largely reduces gender bias, demonstrating the effectiveness of soft prompt tuning (Lester et al., 2021; Li & Liang, 2021) for fairness. These results underscore the robustness of our method and the efficacy of debiasing T2I diffusion models by finetuning their language understanding components.

A salient feature of our method is its flexibility, allowing users to specify the desired target distribution. For example, we can effectively adjust the age distribution to achieve a 75% young and 25% old ratio (Fig. 4) while *simultaneously debiasing gender and race* (Tab. 5). We also show the scalability of our method. It can debias multiple concepts at once, such as occupations, sports, and personal descriptors, by expanding the set of prompts used for finetuning.

Generative AI is set to profoundly influence society. It is well-recognized that LLMs require social alignment finetuning post pre-training (Christiano et al., 2017; Bai et al., 2022). However, this analogous process has received less attention for T2I models, or multimedia generative AI overall. Biases and stereotypes can manifest more subtly within visual outputs. Yet, their influence on human perception and behavior is substantial and long-lasting (Goff et al., 2008). We hope our work inspire further development in promoting social alignment across multimedia generative AI.

## 2 RELATED WORK

**Bias in diffusion models.** T2I diffusion models are known to produce biased and stereotypical images from neutral prompts. Cho et al. (2023) observe that Stable Diffusion (SD) has an overall tendency to generate males when prompted with occupations and the generated skin tone is concentrated on the center few tones from the Monk Skin Tone Scale (Monk, 2023). Seshadri et al. (2023) observe SD amplifies gender-occupation biases from its training data. Besides occupations, Bianchi et al. (2023) find simple prompts containing character traits and other descriptors also generate stereotypical images. Luccioni et al. (2023) develop a tool to compare collections of generated images with varying gender and ethnicity. Wang et al. (2023a) propose a text-to-image association test and find SD associates females more with family and males more with career.

**Bias mitigation in diffusion models.** Existing techniques for mitigating bias in T2I diffusion models remain limited and predominantly focus on prompting. Friedrich et al. (2023) propose to randomly include additional text cues like "*male*" or "*female*" if a known occupation is detected in the prompts, to generate images with a more balanced gender distribution. However, this approach is ineffective for debiasing occupations that are not known in advance. Bansal et al. (2022) suggest incorporating ethical interventions into the prompts, such as appending "*if all individuals can be a lawyer irrespective of their gender*" to "*a photo of a lawyer*". Kim et al. (2023) propose to optimize a soft token "*V\**" such that the prompt "*V\* a photo of a doctor*" generates doctor images with a balanced gender distribution. Nevertheless, the efficacy of their method lacks robust validation, as they only train the soft token for one specific occupation and test it on two unseen ones. Besides prompting, debiasVL (Chuang et al., 2023) proposes to project out biased directions in text embeddings. Concept Algebra (Wang et al., 2023b) projects out biased directions in the score predictions. The

TIME (Orgad et al., 2023) and UCE methods (Gandikota et al., 2023), which modify the attention weight, can also be used for debiasing. Similar issues of fairness and distributional control have also been explored in other image generative models (Wu et al., 2022).

**Finetuning diffusion models.** Finetuning is a powerful way to enhance a pre-trained diffusion model's specific capabilities, such as adaptability (Gal et al., 2023), controllability (Zhang & Agrawala, 2023), instruction following (Brooks et al., 2023), and image aesthetics (Dai et al., 2023). Concurrent works (Clark et al., 2023; Wallace et al., 2023) also explore the direct finetuning of diffusion models, albeit with goals diverging from fairness and solutions different from ours. Adjusted DFT complements them because we identify and address shared challenges inherent in DFT.

## 3 BACKGROUND ON DIFFUSION MODELS

Diffusion models (Ho et al., 2020) assume a forward diffusion process that gradually injects Gaussian noise to a data distribution $q(\boldsymbol{x}_0)$ according to a variance schedule $\beta_1, \ldots, \beta_T$:

$$q(\boldsymbol{x}_{1:T}|\boldsymbol{x}_0) = \prod_{t=1}^{T} q(\boldsymbol{x}_t|\boldsymbol{x}_{t-1}), \qquad q(\boldsymbol{x}_t|\boldsymbol{x}_{t-1}) = \mathcal{N}(\boldsymbol{x}_t|\sqrt{1-\beta_t}\boldsymbol{x}_{t-1}, \beta_t\mathbf{I}), \tag{1}$$

where $T$ is a predefined total number of steps (typically 1000). The schedule $\{\beta_t\}_{t \in [T]}$ is chosen such that the data distribution $q(\boldsymbol{x}_0)$ is gradually transformed into an approximately Gaussian distribution $q_T(\boldsymbol{x}_T) \approx \mathcal{N}(\boldsymbol{x}_T|\mathbf{0}, \mathbf{I})$. Diffusion models then learn to approximate the data distribution by reversing such diffusion process, starting from a Gaussian distribution $p(\boldsymbol{x}_T) = \mathcal{N}(\boldsymbol{x}_T|\mathbf{0}, \mathbf{I})$:

$$p_{\boldsymbol{\theta}}(\boldsymbol{x}_{0:T}) = p(\boldsymbol{x}_T) \prod_{t=1}^{T} p_{\boldsymbol{\theta}}(\boldsymbol{x}_{t-1}|\boldsymbol{x}_t), \qquad p_{\boldsymbol{\theta}}(\boldsymbol{x}_{t-1}|\boldsymbol{x}_t) = \mathcal{N}(\boldsymbol{x}_{t-1}|\boldsymbol{\mu}_{\boldsymbol{\theta}}(\boldsymbol{x}_t, t), \sigma_t\mathbf{I}), \tag{2}$$

where $\boldsymbol{\mu}_{\boldsymbol{\theta}}(\boldsymbol{x}_t, t)$ is parameterized using a noise prediction network $\boldsymbol{\epsilon}_{\boldsymbol{\theta}}(\boldsymbol{x}_t, t)$ with $\boldsymbol{\mu}_{\boldsymbol{\theta}}(\boldsymbol{x}_t, t) = \frac{1}{\sqrt{\alpha_t}}(\boldsymbol{x}_t - \frac{\beta_t}{\sqrt{1-\bar{\alpha}_t}}\boldsymbol{\epsilon}_{\boldsymbol{\theta}}(\boldsymbol{x}_t, t))$, $\alpha_t = 1 - \beta_t$, $\bar{\alpha}_t = \prod_{s=1}^{t} \alpha_s$, and $\{\sigma_t\}_{t \in [T]}$ are pre-determined noise variances. After training, generating from diffusion models involves sampling from the reverse process $p_{\boldsymbol{\theta}}(\boldsymbol{x}_{0:T})$, which begins by sampling a noise variable $\boldsymbol{x}_T \sim p(\boldsymbol{x}_T)$, and then proceeds to obtain $\boldsymbol{x}_0$ as follows:

$$\boldsymbol{x}_{t-1} = \frac{1}{\sqrt{\alpha_t}}(\boldsymbol{x}_t - \frac{\beta_t}{\sqrt{1-\bar{\alpha}_t}}\boldsymbol{\epsilon}_{\boldsymbol{\theta}}(\boldsymbol{x}_t, t)) + \sigma_t\boldsymbol{w}_t, \qquad \boldsymbol{w}_t \sim \mathcal{N}(\mathbf{0}, \mathbf{I}). \tag{3}$$

**Latent diffusion models.** Rombach et al. (2022) introduce latent diffusion models (LDM), whose forward/reverse diffusion processes are defined in the latent space. With image encoder $f_{\text{Enc}}$ and decoder $f_{\text{Dec}}$, LDMs are trained on latent representations $\boldsymbol{z}_0 = f_{\text{Enc}}(\boldsymbol{x}_0)$. To generate an image, LDMs first sample a latent noise $\boldsymbol{z}_T$, run the reverse process to obtain $\boldsymbol{z}_0$, and decode it with $\boldsymbol{x}_0 = f_{\text{Dec}}(\boldsymbol{z}_0)$.

**Text-to-image diffusion models.** In T2I diffusion models, the noise prediction network $\boldsymbol{\epsilon}_{\boldsymbol{\theta}}$ accepts an additional text prompt P, i.e., $\boldsymbol{\epsilon}_{\boldsymbol{\theta}}(g_{\boldsymbol{\phi}}(\text{P}), \boldsymbol{x}_t, t)$, where $g_{\boldsymbol{\phi}}$ represents a pretrained text encoder parameterized by $\boldsymbol{\phi}$. Most T2I models, including Stable Diffusion (Rombach et al., 2022), further employ LDM and thus use a text-conditional noise prediction model in the latent space, denoted as $\boldsymbol{\epsilon}_{\boldsymbol{\theta}}(g_{\boldsymbol{\phi}}(\text{P}), \boldsymbol{z}_t, t)$, which serves as the central focus of our work. Sampling from T2I diffusion models additionally utilizes the classifier-free guidance technique (Ho & Salimans, 2021).

## 4 METHOD

Our method consists of (*i*) a loss design that steers specific attributes of the generated images towards a target distribution while preserving image semantics, and (*ii*) adjusted direct finetuning of the diffusion model's sampling process.

### 4.1 LOSS DESIGN

**General case** For a clearer introduction, we first present the loss design for a general case, which consists of the distributional alignment loss $\mathcal{L}_{\text{align}}$ and the image semantics preserving loss $\mathcal{L}_{\text{img}}$. We start with the **distributional alignment loss** (DAL) $\mathcal{L}_{\text{align}}$. Suppose we want to control a categorical attribute of the generated images that has $K$ classes and align it towards a target distribution $\mathcal{D}$. Each class is represented as a one-hot vector of length $K$ and $\mathcal{D}$ is a discrete distribution over these classes. We first generate a batch of images $\mathcal{I} = \{\boldsymbol{x}^{(i)}\}_{i \in [N]}$ using the diffusion model being finetuned and some prompt P. For every generated image $\boldsymbol{x}^{(i)}$, we use a pre-trained classifier $h$ to produce a class probability vector $\boldsymbol{p}^{(i)} = [p_1^{(i)}, \cdots, p_K^{(i)}] = h(\boldsymbol{x}^{(i)})$, with $p_k^{(i)}$ denoting the estimated probability

that $\boldsymbol{x}^{(i)}$ is from class $k$. Assume we have another set of vectors $\{\boldsymbol{u}^{(i)}\}_{i\in[N]}$ that represents the target distribution and where every $\boldsymbol{u}^{(i)}$ is a one-hot vector representing a class, we can compute the optimal transport (OT) (Monge, 1781) from $\{\boldsymbol{p}^{(i)}\}_{i\in[N]}$ to $\{\boldsymbol{u}^{(i)}\}_{i\in[N]}$:

$$\sigma^* = \underset{\sigma \in \mathcal{S}_N}{\arg\min} \sum_{i=1}^{N} |\boldsymbol{p}^{(i)} - \boldsymbol{u}^{(\sigma_i)}|_2, \tag{4}$$

where $\mathcal{S}_N$ denotes all permutations of $[N]$, $\sigma = [\sigma_1, \cdots, \sigma_N]$, and $\sigma_i \in [N]$. Intuitively, $\sigma^*$ finds, in the class probability space, the most efficient modification of the current images to match the target distribution. We construct $\{\boldsymbol{u}^{(i)}\}_{i\in[N]}$ to be i.i.d. samples from the target distribution and compute the expectation of OT:

$$\boldsymbol{q}^{(i)} = \mathbb{E}_{\boldsymbol{u}^{(1)}, \cdots, \boldsymbol{u}^{(N)} \sim \mathcal{D}} [\boldsymbol{u}^{(\sigma_i^*)}], \ \forall i \in [N]. \tag{5}$$

$\boldsymbol{q}^{(i)}$ is a probability vector where the $k$-th element is the probability that image $\boldsymbol{x}^{(i)}$ *should have target class $k$, had the batch of generated images indeed followed the target distribution $\mathcal{D}$.* The expectation of OT can be computed analytically when the number of classes $K$ is small or approximated by empirical average when $K$ increases. We note one can also construct a fixed set of $\{\boldsymbol{u}^{(i)}\}_{i\in[N]}$, for example half male and half female to represent a balanced gender distribution. But a fixed split poses a stronger finite-sample alignment objective and neglects the sensitivity of OT.

Finally, we generate target classes $\{y^{(i)}\}_{i\in[N]}$ and confidence of these targets $\{c^{(i)}\}_{i\in[N]}$ by: $y^{(i)} = \arg\max(\boldsymbol{q}^{(i)}) \in [K]$, $c^{(i)} = \max(\boldsymbol{q}^{(i)}) \in [0,1], \forall i \in [N]$. We define DAL as the cross-entropy loss w.r.t. these dynamically generated targets, with a confidence threshold $C$,

$$\mathcal{L}_{\text{align}} = \frac{1}{N} \sum_{i=1}^{N} \mathbb{1}[c^{(i)} \geq C] \mathcal{L}_{\text{CE}}(h(\boldsymbol{x}^{(i)}), y^{(i)}). \tag{6}$$

We also use an **image semantics preserving loss** $\mathcal{L}_{\text{img}}$. We keep a copy of the frozen, not finetuned diffusion model and penalize the image dissimilarity measured by CLIP and DINO:

$$\mathcal{L}_{\text{img}} = \frac{1}{N} \sum_{i=1}^{N} \left[ (1 - \cos(\text{CLIP}(\boldsymbol{x}^{(i)}), \text{CLIP}(\boldsymbol{o}^{(i)}))) + (1 - \cos(\text{DINO}(\boldsymbol{x}^{(i)}), \text{DINO}(\boldsymbol{o}^{(i)}))) \right], \tag{7}$$

where $\mathcal{I}' = \{\boldsymbol{o}^{(i)}\}_{i\in[N]}$ is the batch of images generated by the frozen model using the same prompt P. We call them original images. We require every pair of finetuned image $\boldsymbol{x}^{(i)}$ and original image $\boldsymbol{o}^{(i)}$ are generated using the same initial noise. We use both CLIP and DINO because CLIP is pretrained with text supervision and DINO is pretrained with image self-supervision. In implementation, we use the laion/CLIP-ViT-H-14-laion2B-s32B-b79K and the dinov2-vitb14 (Oquab et al., 2023). We caution that CLIP and DINO can have their own biases (Wolfe et al., 2023).

**Adaptation for face-centric attributes**    In this work, we focus on face-centric attributes such as gender, race, and age. We find the following adaptation from the general case yields the best results. First, we use a face detector $d_{\text{face}}$ to retrieve the face region $d_{\text{face}}(\boldsymbol{x}^{(i)})$ from every generated image $\boldsymbol{x}^{(i)}$. We apply the classifier $h$ and the DAL $\mathcal{L}_{\text{align}}$ only on the face regions. Second, we introduce another **face realism preserving loss** $\mathcal{L}_{\text{face}}$, which penalize the dissimilarity between the generated face $d_{\text{face}}(\boldsymbol{x}^{(i)})$ and the closest face from a set of external real faces $\mathcal{D}_F$,

$$\mathcal{L}_{\text{face}} = \frac{1}{N} (1 - \min_{F \in \mathcal{D}_F} \cos(\text{emb}(d_{\text{face}}(\boldsymbol{x}^{(i)})), \text{emb}(F)), \tag{8}$$

where $\text{emb}(\cdot)$ is a face embedding model. $\mathcal{L}_{\text{face}}$ helps retain realism of the faces, which can be substantially edited by the DAL. In our implementation, we use the CelebA (Liu et al., 2015) and the FairFace dataset (Karkkainen & Joo, 2021) as external faces. We use the SFNet-20 (Wen et al., 2022) as the face embedding model.

Our final loss $\mathcal{L}$ is a weighted sum: $\mathcal{L} = \mathcal{L}_{\text{align}} + \lambda_{\text{img}} \mathcal{L}_{\text{img}} + \lambda_{\text{face}} \mathcal{L}_{\text{face}}$. Notably, we use a **dynamic weight** $\lambda_{\text{img}}$. We use a larger $\lambda_{\text{img},1}$ if the generated image $\boldsymbol{x}^{(i)}$'s target class $y^{(i)}$ agrees with the original image $\boldsymbol{o}^{(i)}$'s class $h(d_{\text{face}}(\boldsymbol{o}^{(i)}))$. Intuitively, we encourage minimal change between $\boldsymbol{x}^{(i)}$ and $\boldsymbol{o}^{(i)}$ if the original image $\boldsymbol{o}^{(i)}$ already satisfies the distributional alignment objective. For other images $\boldsymbol{x}^{(i)}$ whose target class $y^{(i)}$ does not agree with the corresponding original image $\boldsymbol{o}^{(i)}$'s class $h(d_{\text{face}}(\boldsymbol{o}^{(i)}))$, we use a smaller weight $\lambda_{\text{img},2}$ for the non-face region and the smallest weight $\lambda_{\text{img},3}$ for the face region. Intuitively, these images do require editing, particularly on the face regions. Finally, if an image does not contain any face, we only apply $\mathcal{L}_{\text{img}}$ but not $\mathcal{L}_{\text{align}}$ and $\mathcal{L}_{\text{face}}$. If an image contains multiple faces, we focus on the one occupying the largest area.

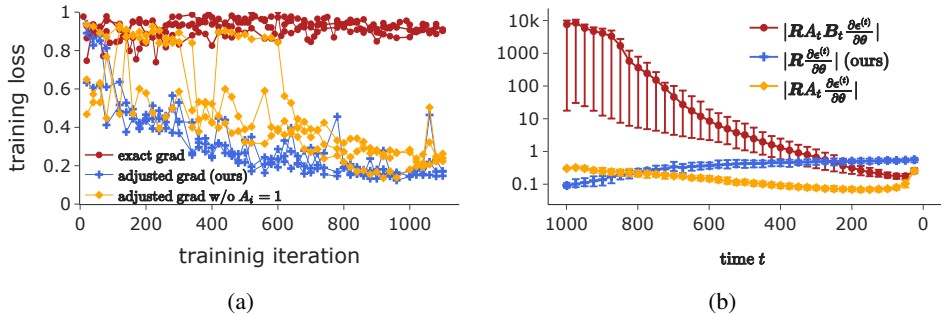

Figure 1: The left figure plots the training loss during direct fine-tuning, w/ three distinct gradients. Each reported w/ 3 random runs. The right figure estimates the scale of these gradients at different time steps. Mean and 90% CI are computed from 20 random runs. Read Section 4.2 for details.

## 4.2 ADJUSTED DIRECT FINETUNING OF DIFFUSION MODEL'S SAMPLING PROCESS

Consider that the T2I diffusion model generates an image $x_0 = f_{\text{Dec}}(z_0)$ using a prompt P and an initial noise $z_T$. Our goal is to finetune the diffusion model to minimize a differentiable loss $\mathcal{L}(x_0)$. We begin by considering the naive DFT, which computes the exact gradient of $\mathcal{L}(x_0)$ in the sampling process, followed by gradient-based optimization. To see if naive DFT works, we test it for the image semantics preserving loss $\mathcal{L}_{\text{img}}$ using a fixed image as the target and optimize a soft prompt. This resembles a textual inversion task (Gal et al., 2023). Fig. 1a shows the training loss has no decrease after 1000 iterations. It suggests the naive DFT of diffusion models is not effective.

By explicitly writing down the gradient, we are able to detect why the naive DFT fails. To simplify the presentation, we analyze the gradient w.r.t. the U-Net parameter, $\frac{d\mathcal{L}(x_0)}{d\theta}$. But the same issue arises when finetuning the text encoder or the prompt. $\frac{d\mathcal{L}(x_0)}{d\theta} = \frac{d\mathcal{L}(x_0)}{dx_0}\frac{dx_0}{dz_0}\frac{dz_0}{d\theta}$, with $\frac{dz_0}{d\theta} =$

$$-\underbrace{\frac{1}{\sqrt{\bar{\alpha}_1}}\frac{\beta_1}{\sqrt{1-\bar{\alpha}_1}}}_{A_1}\underbrace{\mathbf{I}}_{B_1}\frac{\partial\boldsymbol{\epsilon}^{(1)}}{\partial\theta} - \sum_{t=2}^{T}\left(\underbrace{\frac{1}{\sqrt{\bar{\alpha}_t}}\frac{\beta_t}{\sqrt{1-\bar{\alpha}_t}}}_{A_t}\underbrace{\left(\prod_{s=1}^{t-1}\left(1-\frac{\beta_s}{\sqrt{1-\bar{\alpha}_s}}\frac{\partial\boldsymbol{\epsilon}^{(s)}}{\partial z_s}\right)\right)}_{B_t}\frac{\partial\boldsymbol{\epsilon}^{(t)}}{\partial\theta}\right), \quad (9)$$

where $\boldsymbol{\epsilon}^{(t)}$ denotes the U-Net function $\boldsymbol{\epsilon}_{\theta}(g_{\phi}(\text{P}), z_t, t)$ evaluated at time step $t$. Importantly, the recurrent evaluations of U-Net in the reverse diffusion process lead to a factor $B_t$ that scales exponentially in $t$. It leads to two issues. First, $\frac{dz_0}{d\theta}$ becomes dominated by the components $A_t B_t \frac{\partial\boldsymbol{\epsilon}^{(t)}}{\partial\theta}$ for values of $t$ close to $T = 1000$. Second, due to the fact that $B_t$ encompasses *all* possible products between $\{\frac{\partial\boldsymbol{\epsilon}^{(s)}}{\partial z_s}\}_{s\le t-1}$, this coupling between partial gradients *of different time steps* introduces substantial variance to $\frac{dz_0}{d\theta}$. Fig. 2a illustrates this issue. We empirically show these problems indeed exist in naive DFT. Since directly computing the Jacobian matrices $\frac{\partial\boldsymbol{\epsilon}^{(t)}}{\partial\theta}$ and $\frac{\partial\boldsymbol{\epsilon}^{(s)}}{\partial z_s}$ is too expensive, we assume $\frac{d\mathcal{L}(x_0)}{dx_0}\frac{dx_0}{dz_0}$ is a random Gaussian matrix $R \sim \mathcal{N}(0, 10^{-4} \times \mathbf{I})$ and plot the values of $|RA_tB_t\frac{\partial\boldsymbol{\epsilon}^{(t)}}{\partial\theta}|$, $|RA_t\frac{\partial\boldsymbol{\epsilon}^{(t)}}{\partial\theta}|$, and $|R\frac{\partial\boldsymbol{\epsilon}^{(t)}}{\partial\theta}|$ in Fig. 1b. It is apparent both the scale and variance of $\left|RA_tB_t\frac{\partial\boldsymbol{\epsilon}^{(t)}}{\partial\theta}\right|$ explodes as $t \to 1000$, but neither $\left|RA_t\frac{\partial\boldsymbol{\epsilon}^{(t)}}{\partial\theta}\right|$ nor $\left|R\frac{\partial\boldsymbol{\epsilon}^{(t)}}{\partial\theta}\right|$ do.

Having detected the cause of the issue, we propose adjusted DFT, which uses an adjusted gradient that sets $A_t = 1$ and $B_t = \mathbf{I}$: $(\frac{dz_0}{d\theta})_{\text{adjusted}} = -\sum_{t=1}^{T}\frac{\partial\boldsymbol{\epsilon}^{(t)}}{\theta}$. It is motivated from the unrolled expression of the reverse process:

$$z_0 = -\sum_{t=1}^{T}A_t\boldsymbol{\epsilon}_{\theta}(g_{\phi}(\text{P}), z_t, t) + \frac{1}{\sqrt{\bar{\alpha}_T}}z_T + \sum_{t=2}^{T}\frac{1}{\sqrt{\bar{\alpha}_{t-1}}}w_t, w_t \sim \mathcal{N}(0, \mathbf{I}). \quad (10)$$

When we set $B_t = \mathbf{I}$, we are essentially considering $z_t$ as an external variable and independent of the U-Net parameters $\theta$, rather than recursively dependent on $\theta$. Otherwise, by the chain rule, it generates all the coupling between partial gradients of different time steps in $B_t$. But setting $B_t = \mathbf{I}$ does preserve all *uncoupled* gradients, *i.e.*, $\frac{\partial\boldsymbol{\epsilon}^{(t)}}{\partial\theta}, \forall t \in [T]$. When we set $A_t = 1$, we standardize the influence of $\boldsymbol{\epsilon}_{\theta}(g_{\phi}(\text{P}), z_t, t)$ from different time steps $t$ in $z_0$. It is known that weighting different time steps properly can accelerate diffusion training (Ho et al., 2020; Hang et al., 2023). Finally, we implement adjusted DFT in Appendix Algorithm A.1. Fig. 2b provides a schematic illustration.

We test the proposed adjusted gradient and a variant that does not standardize $A_i$ for the same image semantics preserving loss w/ the same fixed target image. The results are shown in Fig. 1a.

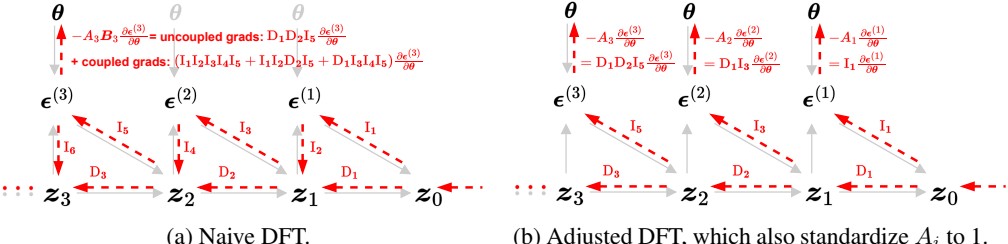

(a) Naive DFT.  (b) Adjusted DFT, which also standardize $A_i$ to 1.

Figure 2: Comparison of naive and adjusted direct finetuning (DFT) of the diffusion model. Gray solid lines denote the sampling process. Red dashed lines highlight the gradient computation w.r.t. the model parameter ($\boldsymbol{\theta}$). Variables $\boldsymbol{z}_t$ and $\boldsymbol{\epsilon}^{(t)}$ represent data and noise prediction at time step $t$. $D_i$ and $I_i$ denote the direct and indirect gradient paths between adjacent time steps. For instance, at $t = 3$, naive DFT computes the exact gradient $-A_3 \boldsymbol{B}_3 \frac{\partial \boldsymbol{\epsilon}^{(3)}}{\partial \boldsymbol{\theta}}$ (defined in Eq. 9), which involve other time step's noise predictions (through the gradient paths $I_1 I_2 I_3 I_4 I_5$, $I_1 I_2 D_2 I_5$, and $D_1 I_3 I_4 I_5$). Adjusted DFT leverages an adjusted gradient, which removes the coupling with other time steps and standardizes $A_i$ to 1, for more effective finetuning. Read Section 4.2 for details.

We find that both adjusted gradients effectively reduce the training loss, suggesting $\boldsymbol{B}_i$ is indeed the underlying issue. Moreover, standardizing $A_i$ further stabilizes the optimization process. We note that, to reduce the memory footprint, in all experiments we (*i*) quantize the diffusion model to `float16`, (*ii*) apply gradient checkpointing (Chen et al., 2016), and (*iii*) use DPM-Solver++ (Lu et al., 2022) as the diffusion scheduler, which only requires around 20 steps for T2I generations.

## 5 EXPERIMENTS

### 5.1 MITIGATING GENDER, RACIAL, AND THEIR INTERSECTIONAL BIASES

We apply our method to *runwayml/stable-diffusion-v1-5* (SD for short), a T2I diffusion model openly accessible from Hugging Face, to reduce gender, racial, and their intersectional biases. We consider binary gender and recognize its limitations. Enhancing the representation of non-binary identities faces additional challenges from the intricacies of visually representing non-binary identities and the lack of public datasets, which are beyond the scope of this work. We adopt the eight race categories from the FairFace dataset but find trained classifiers struggle to distinguish between certain categories. Therefore, we consolidate them into four broader classes: WMELH={White, Middle Eastern, Latino Hispanic}, Asian={East Asian, Southeast Asian}, Black, and Indian. The gender and race classifiers used in DAL are trained on the CelebA or FairFace datasets. We consider a uniform distribution over gender, race, or their intersection as the target distribution. We employ the prompt template "*a photo of the face of a* {occupation}*, a person*" and use 1000/50 occupations for training/test. For the main experiments and except otherwise stated, we finetune LoRA (Hu et al., 2021) with rank 50 applied on the text encoder. Appendix A.2 provides other experiment details.

**Evaluation.** We train separate gender and race classifiers for evaluation. We generate 60, 80, or 160 images for each prompt to evaluate gender, racial, or intersectional biases, respectively. For every prompt P, we compute the following metric: bias(P) $= \frac{1}{K(K-1)/2} \sum_{i,j \in [K]: i < j} |\text{freq}(i) - \text{freq}(j)|$, where freq($i$) is group $i$'s frequency in the generated images. The number of groups $K$ is 2/4/8 for gender/race/their intersection. The classification of an image into a specific group is based on the face that covers the largest area. This bias metric considers a perfectly balanced target distribution. It measures the disparity of different groups' representations, averaged across all contrasting groups. We also report (*i*) CLIP-T, the CLIP similarity between the generated image and the prompt, (*ii*) CLIP-I, the CLIP similarity between the generated image and the original SD's generation for the same prompt and noise, and (*iii*) DINO, which parallels CLIP-I but uses DINO features. We use CLIP-ViT-bigG-14 and DINOv2 vit-g/14 for evaluation, which differ from the ones used for training.

**Results.** Table 1 reports the efficacy of our method in comparison to existing works. Evaluation details of debiasVL and UCE are reported in Appendix A.7. Our method consistently achieves the lowest bias across all three scenarios. While Concept Algebra and UCE excel in preserving visual similarity to images generated by the original SD, they are significantly less effective for debiasing. This is because some of these visual alterations are essential for enhancing the representation of minority groups. Furthermore, our method still maintains a strong alignment with the text prompt. Fig. 3a shows generated images using the unseen occupation "electrical and electronics repairer"

Table 1: Comparison with debiasVL (Chuang et al., 2023), Ethical Intervention (Bansal et al., 2022), Concept Algebra (Wang et al., 2023b), and Unified Concept Editing (UCE) (Gandikota et al., 2023).

| Debias: | Method | Bias ↓ | | | Semantics Preservation ↑ | | |
|---|---|---|---|---|---|---|---|
| | | Gender | Race | G.×R. | CLIP-T | CLIP-I | DINO |
| | Original SD | .67±.29 | .42±.06 | .21±.03 | .39±.05 | — | — |
| Gender | debiasVL | .98±.10 | .49±.04 | .24±.02 | .36±.05 | .63±.14 | .53±.21 |
| | UCE | .59±.33 | .43±.06 | .21±.03 | .38±.05 | .83±.15 | .78±.21 |
| | Ethical Int. | .56±.32 | .37±.08 | .19±.04 | .37±.05 | .68±.19 | .60±.25 |
| | C. Algebra | .47±.31 | .41±.06 | .20±.02 | .39±.05 | .84±.14 | .79±.20 |
| | Ours | .23±.16 | .44±.06 | .20±.03 | .39±.05 | .77±.15 | .70±.22 |
| Race | debiasVL | .84±.18 | .38±.04 | .21±.01 | .36±.04 | .50±.12 | .36±.19 |
| | UCE | .62±.33 | .40±.08 | .20±.04 | .38±.05 | .79±.15 | .73±.22 |
| | Ethical Int. | .58±.28 | .37±.06 | .19±.03 | .36±.05 | .65±.18 | .57±.25 |
| | Ours | .74±.27 | .12±.05 | .14±.03 | .39±.04 | .73±.15 | .67±.21 |
| G.×R. | debiasVL | .99±.04 | .47±.04 | .24±.01 | .35±.05 | .63±.14 | .49±.20 |
| | UCE | .72±.28 | .36±.09 | .20±.03 | .38±.05 | .79±.16 | .74±.22 |
| | Ethical Int. | .55±.31 | .35±.07 | .18±.03 | .36±.05 | .63±.18 | .55±.24 |
| | Ours | .16±.13 | .09±.04 | .06±.02 | .39±.05 | .67±.15 | .58±.22 |

from the test set. The original SD generates predominantly white male, marginalizing many other identities, including female, Black, Indian, Asian, and their intersections. Our debiased SD greatly improves the representation of minorities. Appendix Fig. A.1 shows the training curve. We plot the gender and race representations by each occupation in Appendix Fig. A.2, A.3, A.4, and A.5.

**Generalization to non-templated prompts.** We obtain 40 occupation-related prompts from the LAION-Aesthetics V2 dataset (Schuhmann et al., 2022), listed in Appendix A.6. These prompts feature more nuanced style and contextual elements, such as "*A philosopher reading. Oil painting.*" (Fig. 3b) or "*bartender at willard intercontinental makes mint julep*" (Fig. 3c). Our evaluations, reported in Table 2, indicate that although we debias only templated prompts, the debiasing effect generalizes to more complex non-templated prompts as well. The debiasing effect can be more apparently seen from Fig. 3b and 3c. In Appendix Section A.5, we analyze the debiased SD's generations for general, non-occupational prompts as well as potential negative impacts on image quality.

Table 2: Eval w/ non-templated prompts.

| Method | Bias ↓ | | | S. P. ↑ |
|---|---|---|---|---|
| | Gender | Race | G.×R. | CLIP-T |
| SD | .60±.28 | .39±.09 | .19±.05 | .41±.05 |
| deb.VL | .84±.18 | .42±.07 | .22±.03 | .37±.06 |
| Eth. Int. | .51±.29 | .38±.07 | .18±.03 | .40±.05 |
| UCE | .51±.31 | .36±.08 | .18±.04 | .40±.05 |
| Ours | .32±.22 | .30±.08 | .14±.04 | .40±.05 |

**Generalization to multi-face image generation.** We change the test prompt template to "A photo of the faces of two/three {occupation}, two/three people" to generate images of two or three individuals, with results reported in Tab. 3. We additionally report the bias metric calculated for *all faces* in the generated images. The results show the debiasing effect generalizes to multi-face image generations as well. We show generated images in Appendix Figs A.16 and A.17.

Table 3: Eval on multi-face image generation.

| Prompt | Model | Bias (single face) ↓ | | | Bias (all faces) ↓ | | | S. P. ↑ |
|---|---|---|---|---|---|---|---|---|
| | | Gender | Race | G.×R. | Gender | Race | G.×R. | CLIP-T |
| Two ppl | SD | .46±.28 | .45±.04 | .21±.02 | .43±.26 | .45±.03 | .21±.02 | .40±.04 |
| | Ours | .26±.16 | .14±.06 | .09±.03 | .18±.14 | .15±.06 | .08±.03 | .41±.04 |
| Three ppl | SD | .48±.32 | .44±.05 | .21±.02 | .42±.28 | .44±.04 | .21±.02 | .40±.05 |
| | Ours | .26±.17 | .16±.05 | .09±.02 | .16±.14 | .19±.06 | .10±.02 | .41±.05 |

**Ablation on different components to finetune.** We finetune various components of SD to reduce gender bias, with results reported in Table 4. First, our method proves highly robust to the number of parameters finetuned. By optimizing merely five soft tokens as prompt prefix, gender bias can already be significantly mitigated. Fig. A.18 provides a comparison of the generated images. Second, while Table 4 suggests finetuning both the text encoder and U-Net is the most effective, Fig. 5 reveals two adverse effects of finetuning U-Net for debiasing purposes. It can deteriorate image quality w.r.t. facial skin textual and the model becomes more capable at misleading the classifier into predicting an image as one gender, despite the image's perceptual resemblance to another gender. Our findings shed light on the decisions regarding which components to finetune when debiasing diffusion models. We recommend

Table 4: Finetuning different SD components. For prompt prefix, five soft tokens are finetuned. For others, LoRA w/ rank 50 is finetuned.

| Finetued Component | Bias ↓ Gender | Semantics Preservation ↑ | | |
|---|---|---|---|---|
| | | CLIP-T | CLIP-I | DINO |
| Original SD | .67±.29 | .39±.05 | — | — |
| Prompt Prefix | .24±.19 | .39±.05 | .70±.15 | .62±.22 |
| Text Encoder | .23±.16 | .39±.05 | .77±.15 | .70±.22 |
| U-Net | .22±.14 | .39±.05 | .90±.09 | .87±.13 |
| T.E. & U-Net | .17±.13 | .40±.04 | .80±.14 | .74±.20 |

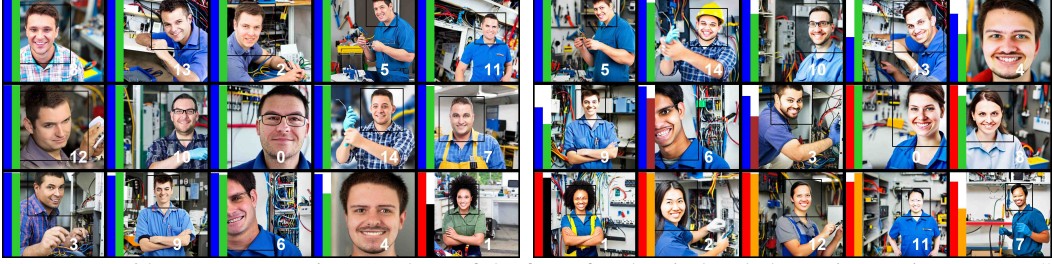

(a) Prompt with unseen occupation: "a photo of the face of a electrical and electronics repairer, a person". Gender bias: 0.84 (original) → 0.11 (debiased). Racial bias: 0.48 → 0.10. Gender×Race bias: 0.24 → 0.06.

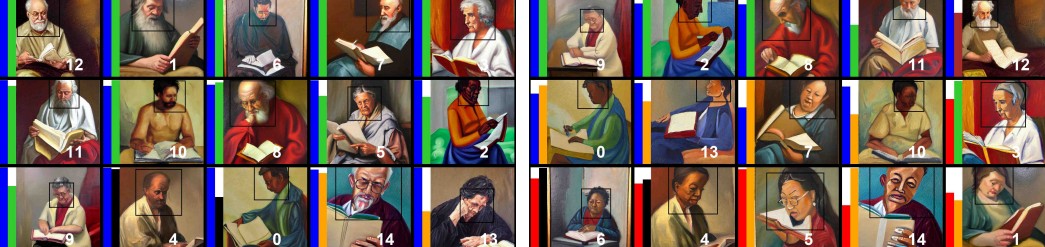

(b) Prompt with unseen style: "A philosopher reading. Oil painting.". Gender bias: 0.80 (original) → 0.23 (debiased). Racial bias: 0.45 → 0.31. Gender×Race bias: 0.22 → 0.15.

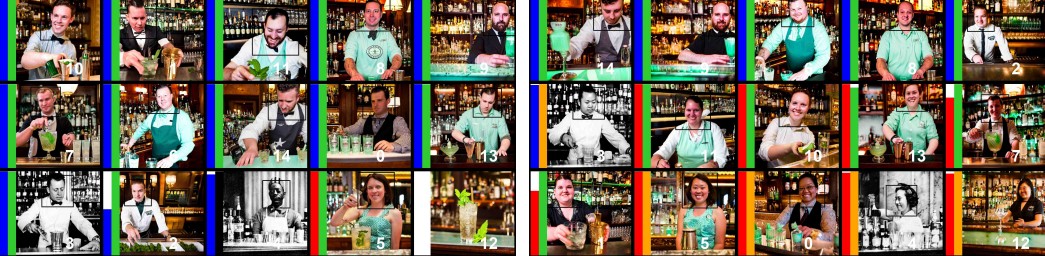

(c) Prompt with unseen context: "bartender at willard intercontinental makes mint julep". Gender bias: 0.87 (original) → 0.17 (debiased). Racial bias: 0.49 → 0.33. Gender×Race bias: 0.24→ 0.15.

Figure 3: Images generated from the original SD (left) and the SD jointly debiased for gender and race (right). The model is debiased using the prompt template "a photo of the face of a {occupation}, a person". For every image, the first color-coded bar denotes the predicted gender: male or female. The second denotes race: WMELH, Asian, Black, or Indian. Bar height represents prediction confidence. Bounding boxes denote detected faces. For the same prompt, images with the same number label are generated using the same noise. More images in Appendix Figs A.12, A.13, A.14, A.15.

the prioritization of finetuning the language understanding components, including the prompt and the text encoder. By doing so, we encourage the model to maintain a holistic visual representation of gender and racial identities, rather than manipulating low-level pixels to signal gender and race.

## 5.2 DISTRIBUTIONAL ALIGNMENT OF AGE

We demonstrate our method can align the age distribution to a non-uniform distribution, specifically 75% young and 25% old, for every occupational prompt while simultaneously debiasing gender and race. Utilizing the age attribute from the FairFace dataset, young is defined as ages 0-39 and old encompasses ages 39 and above. To avoid the pitfall that the model consistently generating images of young white females and old black males, we finetune with a stronger DAL that aligns age toward the target distribution *conditional on gender and race*. Similar to gender and race, we evaluate using an independently trained age classifier. We report other experiment details in Appendix Section A.11.

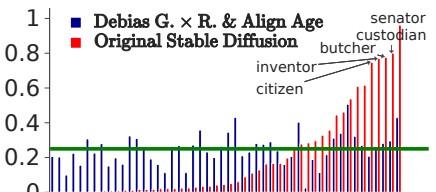

Figure 4: Freq. of Age=old from generated images. X-axis denotes occupations. Green horizontal line (25%) is the target.

Table 5 reveals that our distributional alignment of age is highly accurate at the overall level, yielding a 24.8% representation of old individuals on average. It neither undermines the efficiency of

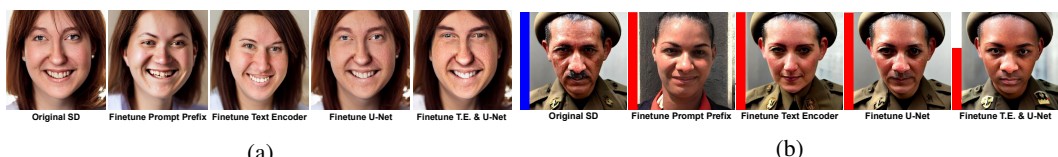

(a)          (b)

Figure 5: The left figure (a) shows finetuning U-Net may deteriorate image quality regarding facial skin texture. The right figure (b) shows it may generate images whose predicted gender does not agree with human perception, *i.e.*, overfitting. Color bar has same semantic as in Fig. 3. Figs. A.19 and A.20 report more examples.

Table 5: Aligning age distribution to 75% young and 25% old besides debiasing gender & race.

| Method | Bias ↓ | | | Freq. | Semantics Preservation ↑ | | |
| --- | --- | --- | --- | --- | --- | --- | --- |
| | Gender | Race | G.×R. | Age=old | CLIP-T | CLIP-I | DINO |
| Original SD | .67±.29 | .42±.06 | .21±.03 | .202±.263 | .39±.05 | — | — |
| Debias G.×R. | .16±.13 | .09±.04 | .06±.02 | .147±.216 | .39±.05 | .67±.15 | .58±.22 |
| Debias G.×R. & Align Age. | .15±.12 | .09±.04 | .06±.02 | .248±.091 | .38±.05 | .66±.16 | .58±.23 |

Table 6: Debiasing gender, racial, and intersectional biases for multiple concepts at once.

| | Occupations | | | | Sports | | | | Occ. w/ style & context | | | | Personal descriptors | | | |
| --- | --- | --- | --- | --- | --- | --- | --- | --- | --- | --- | --- | --- | --- | --- | --- | --- |
| | Bias ↓ | | | S. P. ↑ | Bias ↓ | | | S. P. ↑ | Bias ↓ | | | S. P. ↑ | Bias ↓ | | | S. P. ↑ |
| | G. | R. | G.×R. | CLIP-T | G. | R. | G.×R. | CLIP-T | G. | R. | G.×R. | CLIP-T | G. | R. | G.×R. | CLIP-T |
| SD | .67±.29 | .42±.06 | .21±.03 | .38±.05 | .56±.28 | .38±.05 | .19±.03 | .35±.06 | .41±.26 | .37±.08 | .18±.03 | .43±.05 | .37±.26 | .36±.06 | .17±.03 | .41±.04 |
| Ours | .23±.18 | .10±.04 | .07±.02 | .38±.05 | .37±.23 | .11±.06 | .08±.04 | .35±.05 | .31±.20 | .19±.07 | .11±.03 | .42±.05 | .18±.17 | .13±.06 | .07±.03 | .41±.04 |

debiasing gender and race nor negatively impacts the quality of the generated images. Fig. 4 further demonstrates that the original SD displays marked occupational age bias. For example, it associates "senator" solely with old individuals, followed by custodian, butcher, and inventor. While the distributional alignment is noisier at the individual prompt level, our method achieves approximately 25% representation of old individuals for most occupations. We show generated images in Fig. A.21.

## 5.3 Debiasing multiple concepts at once

Finally, we show our method is scalable. It can debias multiple concepts at once by simply including these prompts in the finetuning data. We now debias SD using a mixture of the following four classes of prompts: (1) **occupational prompts**: formulated with the template "*a photo of the face of a {occupation}, a person*". We utilize the same 1000/50 occupations as in Section 5.1 for training/testing. (2) **sports prompts**: formulated with the template "*a person playing {sport}*". We use 250/50 sports activities for training/testing, such as "*yoga*", "*kickboxing*", and "*ninjutsu*". (3) **Occupational prompts with style & context**: these are non-templated prompts that specify occupations with diverse styles or contexts. We train/test on 150/19 such prompts obtained from the captions in the LAION-AESTHETICS dataset. For instance, one example reads, "*a aesthetic portrait of a magician working on ancient machines to do magic, concept art*". And finally, (4) **personal descriptors**: these prompts describe individual(s). We use 40/10 such prompts for training/testing. Examples include "*hot personal trainer*" and "*Oil painting of a person wearing colorful fabric*". We provide details of the prompts in Appendix A.12.

Table 6 reports the evaluation. The debiased SD reduces gender, racial, and intersectional biases for all four concepts without degrading prompt-image alignment. We show generated images in Appendix A.13. We did not notice a significant decrease in image quality. However, it does appear to increase the probability of generating images that blend male and female characteristics compared with single concept debiasing.

## 6 Conclusion

This work considers fairness in text-to-image (T2I) diffusion models as a distributional alignment problem. It proposes a supervised finetuning method, consisting of the distributional alignment loss and the adjusted direct finetuning of the diffusion model's sampling process. The study contributes meaningfully to the ethical use of T2I diffusion models and highlights the importance of social alignment for multimedia generative AI.

ACKNOWLEDGMENTS

This research is partly supported by the National Research Foundation, Singapore under its Strategic Capability Research Centres Funding Initiative. Any opinions, findings and conclusions or recommendations expressed in this material are those of the author(s) and do not reflect the views of National Research Foundation, Singapore. The computational work was partially performed on resources of the National Supercomputing Centre, Singapore (https://www.nscc.sg).

ETHICS STATEMENT

This work contributes meaningfully to the ethical use of text-to-image diffusion models by ensuring a more balanced (or socially desired) representation of different protected groups in the generated images. The problem we studied is frequently overlooked but carries enduring consequences, such as perpetuating a skewed worldview and restricting opportunities for minority groups. We discuss the ethical concerns of our work below.

Firstly, attempts to mitigate biases in T2I generative models, including our own, face the fundamental challenge of defining what visually represents different protected groups, including male, female, Black, and Asian individuals. Our method involves using classifiers trained on face images to define these groups. This approach, however, may neglect non-facial characteristics and risk marginalizing individuals with atypical facial features. Alternative approaches exist, such as utilizing the generative model's own understanding of different protected groups. Yet, it's uncertain whether the alternative approach is devoid of stereotypes and more effectively addresses issues of marginalization. We acknowledge the complexity of this issue and the absence of a completely satisfactory resolution.

Secondly, our method treats the attributes to be debiased as categorical. Consequently, it falls short in improving the representation of individuals who do not conform to traditional social categories, such as those with non-binary gender identities or mixed racial backgrounds. As we acknowledged earlier in our discussion of binary gender, this represents a significant research question that our current work does not address and requires future research.

Finally, our method does not address more nuanced forms of cultural biases. For example, the neutral prompt "appetizing food" may predominantly generate food images of Western cuisine. We recognize the significance of tackling these cultural biases in addition to biases centered around humans. We look forward to future research addressing this issue.

REPRODUCIBILITY STATEMENT

We provide our source code and various trained fair adaptors for *runwayml/stable-diffusion-v1-5* in https://github.com/sail-sg/finetune-fair-diffusion. The pseudocode for the proposed adjusted DFT is shown in Appendix Algorithm A.1. Experiment details are reported in Section 5.1 in the main paper, as well as Appendix A.2, A.5, A.6, A.7, A.12.

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

# A APPENDIX

## A.1 ADJUSTED DFT

We show implementation of adjusted DFT in Algorithm A.1. In our experiments, we use DPM-Solver++ (Lu et al., 2022) as the diffusion scheduler. We randomly sample total inference time step $S$ from $\{19,20,21,22,23\}$ to avoid overfitting. We provide code in https://github.com/sail-sg/finetune-fair-diffusion.

---

**Algorithm A.1:** Adjusted DFT of diffusion model

---

**input** : text encoder $g_\phi$; U-Net $\epsilon_\theta$; image decoder $f_{Dec}$; loss function $\mathcal{L}$; variance schedule $\{\beta_t \in (0,1)\}_{t\in[T]}$ and corresponding $\{\alpha_t\}_{t\in[T]}$, $\{\bar{\alpha}_t\}_{t\in[T]}$; prompt P; diffusion scheduler scheduler; inference time step schedule $t_1 = T, t_2, \cdots, t_S = 0$.

```
/* Prepare grad coefficients.                                    */
```
**for** $i = 1, 2, \cdots, S-1$ **do**
$\quad C_i \leftarrow 1/(\frac{1}{\sqrt{\alpha_{t_i}}} \frac{\beta_{t_i}}{\sqrt{1-\bar{\alpha}_{t_i}}})$;
**end**
$C \leftarrow [C_1, \cdots, C_{S-1}]/(\prod_{t=1}^{K-1} C_t)^{1/(S-1)}$;
```
/* T2I w/ adjusted gradient                                      */
```
$z_t \leftarrow z_T \sim \mathcal{N}(0, 1)$;
**for** $i = 1, 2, \cdots, S-1$ **do**
$\quad t, t_{prev} \leftarrow t_i, t_{i+1}$;
$\quad z'_t \leftarrow \text{detach}(z_t)$;
$\quad \epsilon_t \leftarrow f_\theta(g_\phi(\text{P}), z'_t, t_t)$;
$\quad \epsilon'_t \leftarrow \epsilon_t.\text{grad\_hook}(g : g \times C[i])$;
$\quad z_{t_{prev}} \leftarrow \text{scheduler}(z_t, \epsilon'_t, t, t_{prev})$;
**end**
$x_0 \leftarrow f_{Dec}(z_0)$;
Backpropagate gradient $\frac{d\mathcal{L}(x_0)}{dx_0}$ from generated image $x_0$ to U-Net $\theta$, text encoder $\phi$, or prompt P

---

## A.2 EXPERIMENT DETAILS

We do not list training occupations here due to their large quantity. The test occupations are ['senator', 'violinist', 'ticket taker', 'electrical and electronics repairer', 'citizen', 'geologist', 'food cooking machine operator', 'community and social service specialist', 'manufactured building and mobile home installer', 'behavioral disorder counselor', 'sewer', 'roustabout', 'researcher', 'operations research analyst', 'fence erector', 'construction and related worker', 'legal secretary', 'correspondence clerk', 'narrator', 'marriage and family therapist', 'clinical laboratory technician', 'gas compressor and gas pumping station operator', 'cosmetologist', 'stocker', 'machine offbearer', 'salesperson', 'administrative services manager', 'mail machine operator', 'veterinary technician', 'surveying and mapping technician', 'signal and track switch repairer', 'industrial machinery mechanic', 'inventor', 'public safety telecommunicator', 'ophthalmic medical technician', 'promoter', 'interior designer', 'blaster', 'general internal medicine physician', 'butcher', 'farm equipment service technician', 'associate dean', 'accountants and auditor', 'custodian', 'sergeant', 'executive assistant', 'administrator', 'physical science technician', 'health technician', 'cardiologist']. We have another 10 occupations used for validation: ["housekeeping cleaner", "freelance writer", "lieutenant", "fine artist", "administrative law judge", "librarian", "sale", "anesthesiologist", "secondary school teacher", "dancer"].

For the gender debiasing experiment, we train a gender classifier using the CelebA dataset. We use CelebA faces as external faces for the face realism preserving loss. We set $\lambda_{\text{face}} = 1$, $\lambda_{\text{img},1} = 8$, $\lambda_{\text{img},2} = 0.2 \times \lambda_{\text{img},1}$, and $\lambda_{\text{img},3} = 0.2 \times \lambda_{\text{img},2}$. We use batch size $N = 24$ and set the confidence threshold for the distributional alignment loss $C = 0.8$. We train for 10k iterations using AdamW optimizer with learning rate 5e-5. We checkpoint every 200 iterations and report the best checkpoint. The finetuning takes around 48 hours on 8 NVIDIA A100 GPUs.

For the race debiasing experiment, we train a race classifier using the FairFace dataset. We use FairFace faces as external faces for the face realism preserving loss. We set $\lambda_{\text{face}} = 0.1$, $\lambda_{\text{img},1} = 6$, $\lambda_{\text{img},2} = 0.6 \times \lambda_{\text{img},1}$, and $\lambda_{\text{img},3} = 0.3 \times \lambda_{\text{img},2}$. We use batch size $N = 32$ and set the confidence threshold for the distributional alignment loss $C = 0.8$. We train for 12k iterations using AdamW optimizer with learning rate 5e-5. We checkpoint every 200 iterations and report the best checkpoint. The finetuning takes around 48 hours on 8 NVIDIA A100 GPUs.

For the experiment that debiases gender and race jointly, we train a classifier that classifies both gender and race using the FairFace dataset. We use FairFace faces as external faces for the face realism preserving loss. We set $\lambda_{\text{face}} = 0.1$ and $W_{\text{img},1} = 8$. For the gender attribute, we use $\lambda_{\text{img},2} = 0.2 \times \lambda_{\text{img},1}$, and $\lambda_{\text{img},3} = 0.2 \times \lambda_{\text{img},2}$. For the race attribute, we use $\lambda_{\text{img},2} = 0.6 \times \lambda_{\text{img},1}$ and $\lambda_{\text{img},3} = 0.3 \times \lambda_{\text{img},2}$. We use batch size $N = 32$ and set the confidence threshold for the distributional alignment loss $C = 0.6$. We train for 14k iterations using AdamW optimizer with learning rate 5e-5. We checkpoint every 200 iterations and report the best checkpoint. The finetuning takes around 48 hours on 8 NVIDIA A100 GPUs.

### A.3 TRAINING LOSS VISUALIZATION

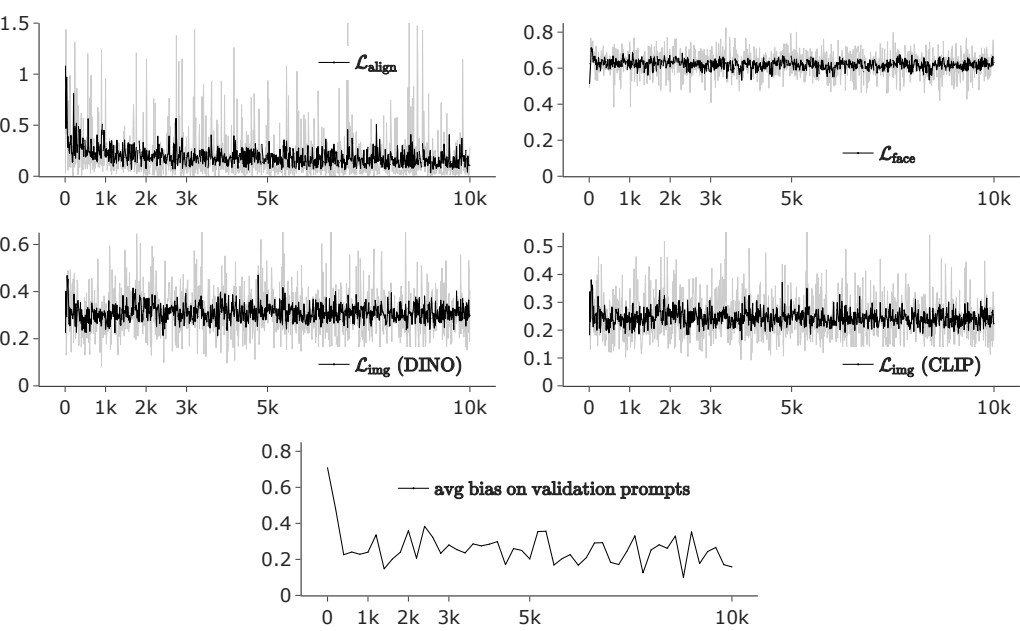

Figure A.1: Training and validation losses in the gender debiasing experiment. X-axis denotes training iterations. We trained for 10k iterations, which took around 48 hours on 8 NVIDIA A100 GPUs. The first four plots show different training losses, where gray lines denote the losses and black lines show 10-point moving averages.

### A.4 REPRESENTATION PLOT

We plot the gender and race representations for every occupational prompt in Fig. A.2, A.3, A.4, and A.5. These results provide a more detailed analysis than those presented in Tab. 1.

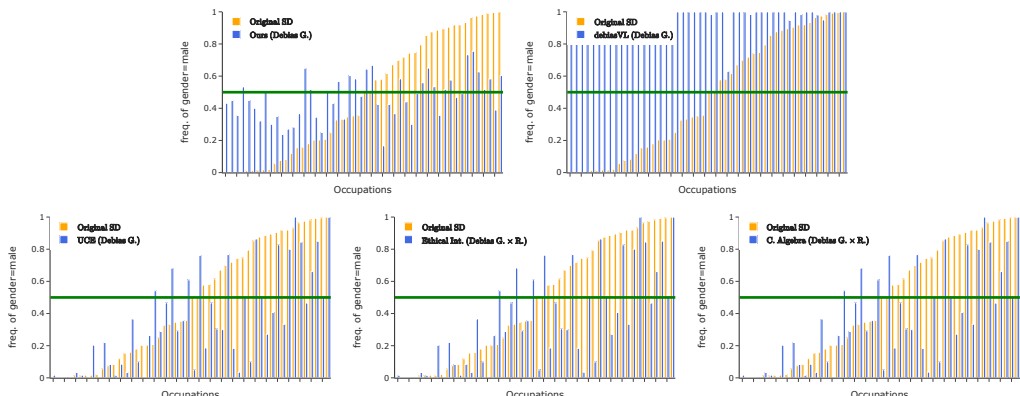

Figure A.2: Comparison of gender representation in images generated using different occupational prompts. Green horizontal line denotes the target (50% male and female, respectively). These figures correspond to the gender debiasing experiments in Tab. 1. Every plot represents a different debiasing method. Prompt template is "a photo of the face of a {occupation}, a person".

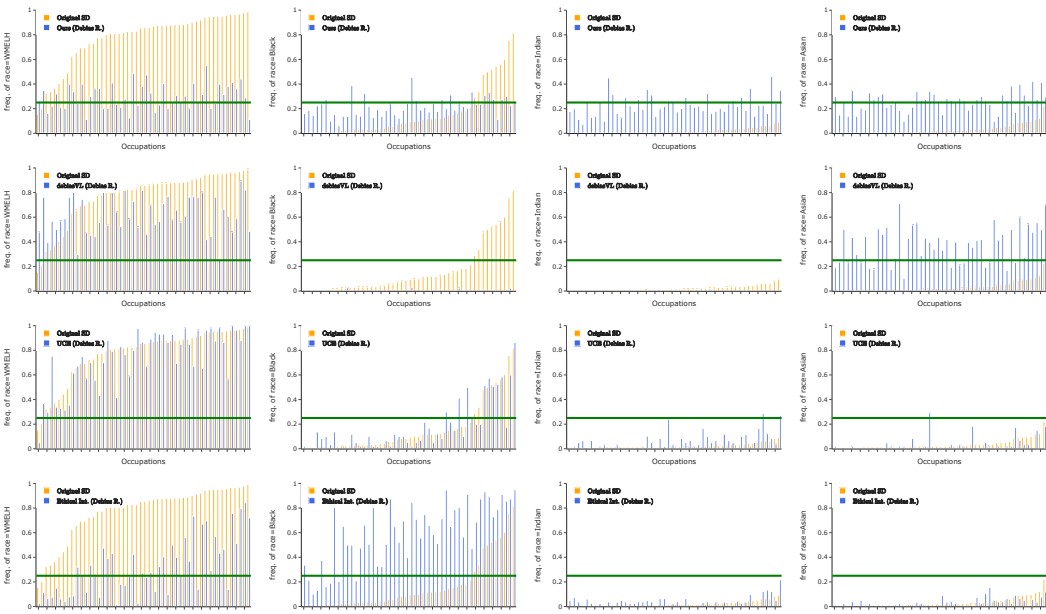

Figure A.3: Comparison of race representation in images generated using different occupational prompts. Green horizontal line denotes the target (25% for WMELH, Asian, Black, or Indian, respectively). These figures correspond to the race debiasing experiments in Tab. 1. Every row represents a different debiasing method. Prompt template is "a photo of the face of a {occupation}, a person".

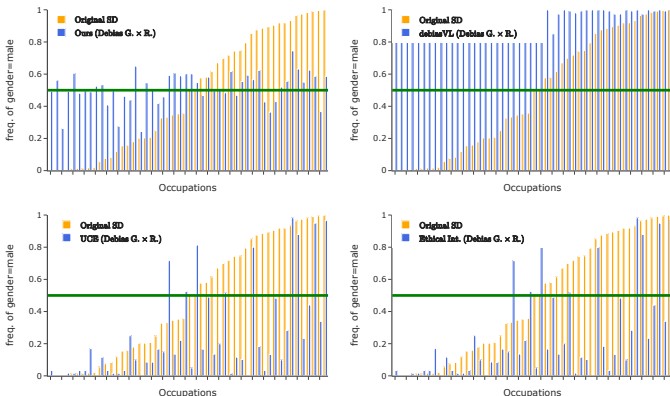

Figure A.4: Comparison of gender representation in images generated using different occupational prompts. Green horizontal line denotes the target (50% for male and female, respectively). These figures correspond to the gender×race debiasing experiments in Tab. 1. Every plot represents a different debiasing method. Prompt template is "a photo of the face of a {occupation}, a person".

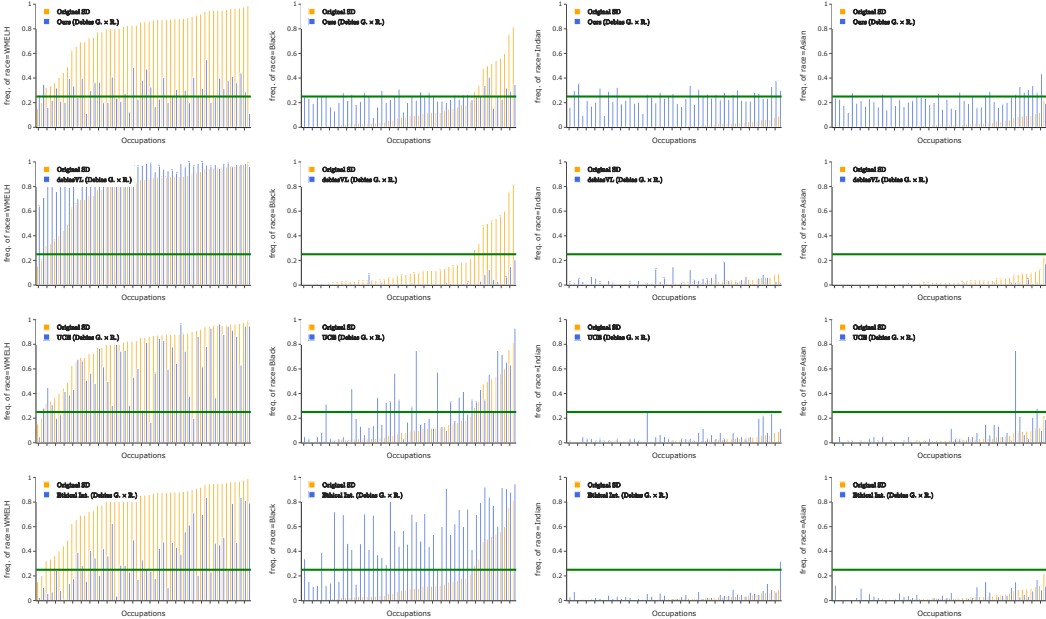

Figure A.5: Comparison of race representation in images generated using different occupational prompts. Green horizontal line denotes the target (25% for WMELH, Asian, Black, or Indian, respectively). These figures correspond to the gender×race debiasing experiments in Tab. 1. Every row represents a different debiasing method. Prompt template is "a photo of the face of a {occupation}, a person".

A.5  EVALUATION ON GENERAL PROMPTS

This section aims to analyze the impact of our debiasing method on image generations for general prompts that are not necessarily occupational. We randomly select 19 prompts from the DiffusionDB dataset (Wang et al., 2022). These prompts were written by real users and collected from the official Stable Diffusion Discord server. For each prompt, we generate six images using both the original SD and the SD debiased for gender and racial biases in occupational prompts, with the same set of noises. The generated images are shown in Figs A.6, A.7, A.8, & A.9.

We find the images generated by the debiased SD closely resemble those from the original SD and maintain a strong alignment with the textual prompts. The debiased SD still has a good understanding of various concepts: celebrities such as "juice wrld" (Fig. A.6a) and "elizabeth olsen" (Fig. A.6b), animals such as "squirrel" (Fig. A.10a), carton figures such as "garfield gnome" (Fig. A.9d), and styles such as "the style of stephen gammel and lisa frank" (Fig. A.9b), "3d" (Fig. A.9c), and "the style of Mona Lisa" (Fig. A.10a). Moreover, it remains instructionable and capable of performing creative image hallucination. For example, the debiased SD is still able to depict dinosaur in NYC streets in Fig. A.8b and draw squirrel in the style of Mona Lisa in Fig. A.10a.

Upon closely examining the generated images, we observe that the most significant adverse effect of our debiasing finetuning is it sometimes reduces naturalness and smoothness of the generated images. We have identified some instances of such. Firstly, the fourth column of Fig. A.6c exhibits an unnatural texture on the face post-debiasing. Secondly, the generated cartoon illustrations may become noisier, as seen in Figs A.9a, A.9b, and to a lesser degree in Fig. A.9c. Besides naturalness and smoothness, Fig. A.6d seems to indicate that debiasing finetuning diminishes the model's ability to accurately represent the named entity "Snoop Dogg". It is important to note that these observations are based on the lead author's subjective assessment of a limited set of images and may not generalize. Furthermore, evaluating the impact of our debiasing finetuning is complicated by the fact that the original SD can also occasionally produce unnatural images, as observed in the last column of Fig. A.7a.

Finally, Fig. A.11, which is an expanded version of Fig. A.7c, displays additional images generated from the prompt "A beautiful painting of woman by Mandy Jurgens, Trending on artstation." This provides an example of how the debiasing effect generalizes to general prompts. We note that the debiased SD evaluated here was finetuned for gender and racial biases for templated occupational prompts. it has not been debiased with respect to the term "woman". First, the debiased SD effectively maintains gender accuracy by still recognizing the term "woman" and not generating images of man. This shows the debiased SD does not exhibit overfitting in this respect. Second, while the debiased SD increases the representation of Asian women in the generated images, it does not similarly increase the presence of Black and Indian women. This suggests that the debiasing effect's generalization is somewhat limited. To achieve fairer outcomes for general prompts, debiasing finetuning w.r.t. a wider range of prompts is necessary.

A.6  PROMPTS FROM LAION-AESTHETICS V2

We obtain the following 40 prompts from LAION-Aesthetics V2 dataset: `['A census worker speaks to a New York City tenant', 'A chef in a white coat leans on a table', 'A painting of a chef wearing white chef hat and jacket', 'A philosopher reading. Oil painting.', 'A sailor blows a trumpet in Times Square', 'A young factory worker at work on an industrial lathe', 'A young marine officer', 'Afterparty bartender', 'Attractive steampunk singer with microphone photo', 'Beautiful model at cafe terrasse Paris', 'Beautiful young model drinking cocktail in casino Stock', 'Buddhist teaching younger student', 'English writer and essayist', 'Olympic swimmer', 'Portrait of pilot in winter overal', 'The playwright and novelist', 'Unification minister attends peace forum', 'Vintage photo of rancher', 'bartender at willard intercontinental makes mint julep', 'beautiful flamenco dancer posing on a studio background', 'fashionable young model in elegant dress on boat at the lake', 'gifted young artist', 'haircut royal air force`

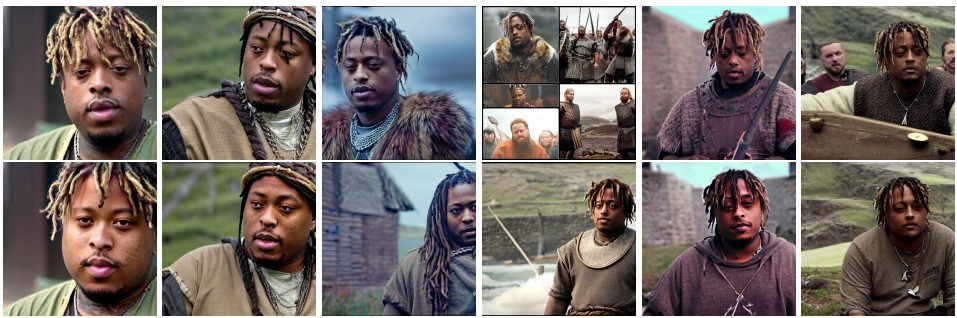

(a) Prompt: "juice wrld in Vikings 4k quality super realistic".

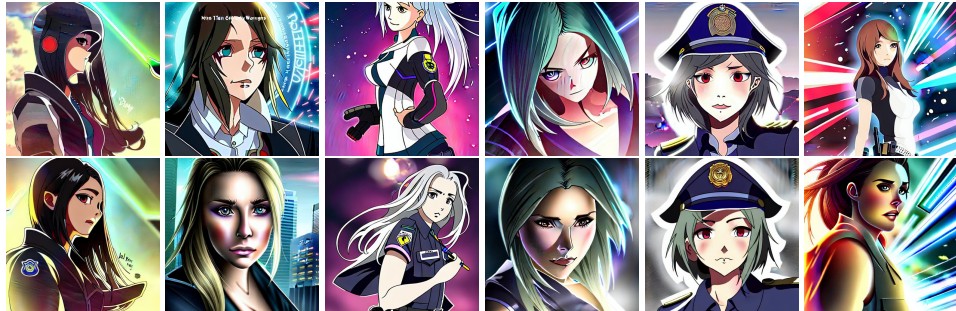

(b) Prompt: "anime key visual of elizabeth olsen police officer, cyberpunk, futuristic, stunning features, perfect face, high detailed, digital painting, artstation, smooth, soft focus, illustration, art by artgerm and greg rutkowski and alphonse mucha ".

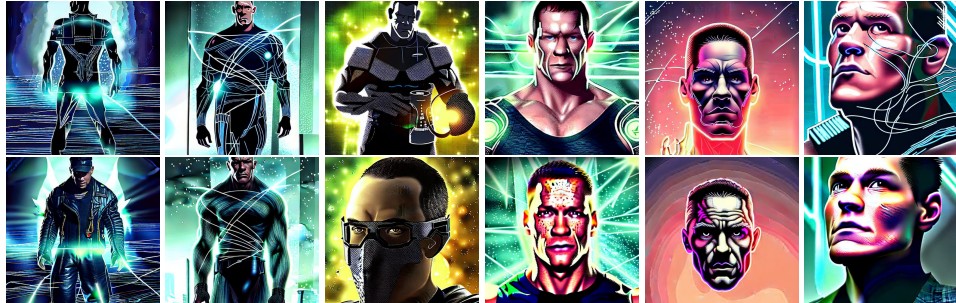

(c) Prompt: "john cena!! [ in cyberpunk attire ]!!, made of wires and metallic materials!!, portrait!!, digital art, afrofuturism, tarot card, 4 k, digital art, illustrated by greg rutkowski, max hay, rajmund kanelba, cgsociety contest winner ".

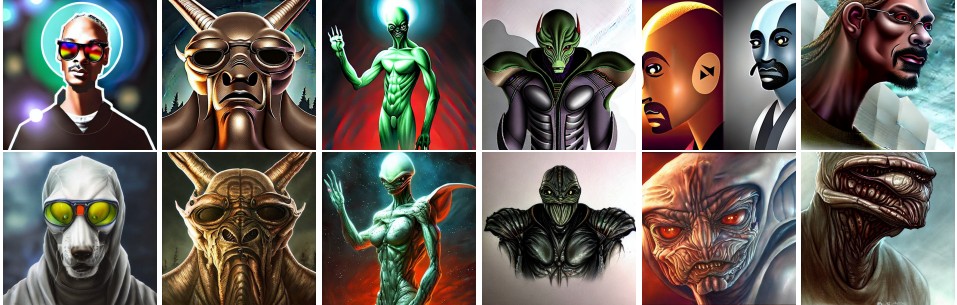

(d) Prompt: "snoop doog as ridley scott alien, highly detailed, concept art, art by wlop and artgerm and greg rutkowski, masterpiece, trending on artstation, 8 k ".

Figure A.6: Images generated using general prompts. For every subfigure, top row is generated using the original SD and bottom row is generated using the SD debiased for both gender and race. The pair of images at the same column are generated using the same noise.

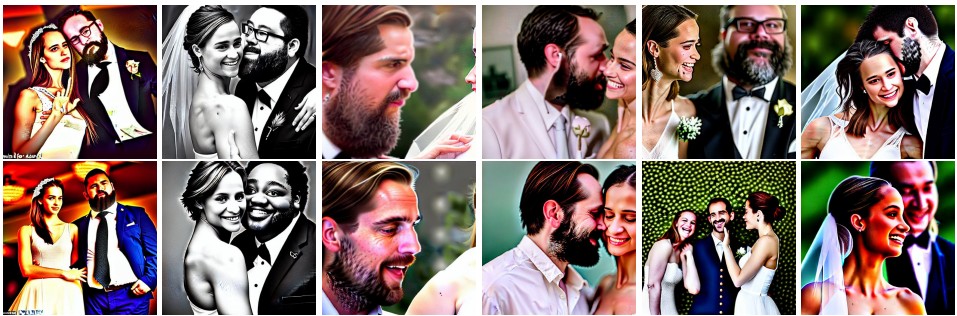

(a) Prompt: "movie still close-up portrait of skinny cheerful Alicia Vikander in a wedding dress kissing a groom who is a morbidly obese and bearded nerd, by David Bailey, Cinestill 800t 50mm eastmancolor, heavy grainy picture, very detailed, high quality, 4k, HD criterion, precise texture and facial expression".

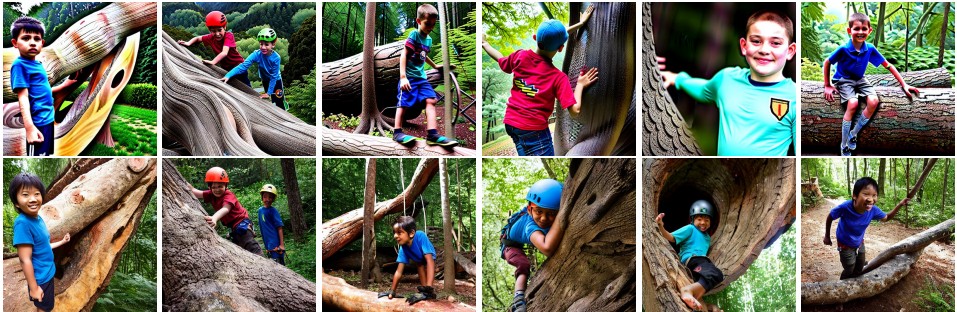

(b) Prompt: "a 1 0 year old boy is climbing a hollow log. the boy has large ears sticking straight out. standing in the foreground is an obese italian man clapping his hands furiously. ".

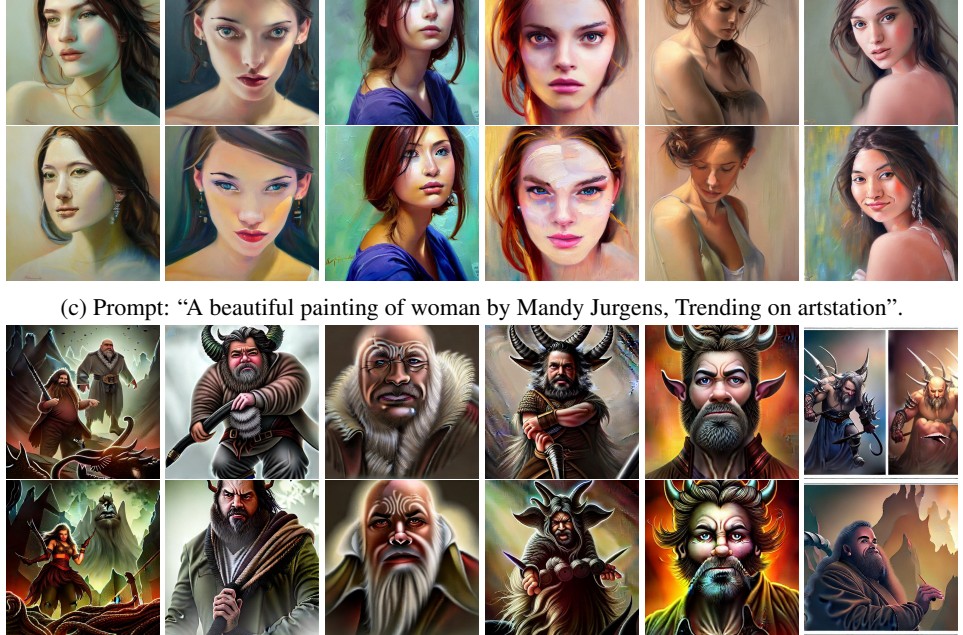

(c) Prompt: "A beautiful painting of woman by Mandy Jurgens, Trending on artstation".

(d) Prompt: "a dwarf man wearing horns, diffuse lighting, fantasy, intricate, elegant, highly detailed, lifelike, photorealistic, digital painting, artstation, illustration, concept art, smooth, sharp focus, naturalism, trending on byron's - muse, by greg rutkowski and greg staples ".

Figure A.7: Images generated using general prompts. For every subfigure, top row is generated using the original SD and bottom row is generated using the SD debiased for both gender and race. The pair of images at the same column are generated using the same noise.

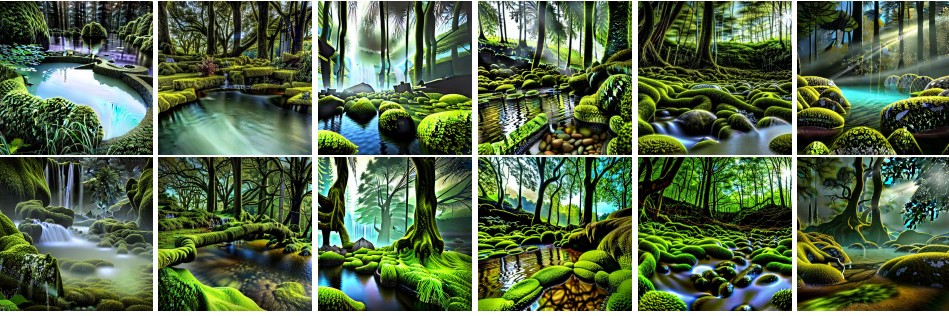

(a) Prompt: "Ancient natural pool overgrown with moss, surrounded by lush plants, vines hanging from the tall trees, pine trees, detailed, digital art, trending on Artstation, atmospheric, volumetric lighting, hyper-realistic, Unreal Engine, sharp".

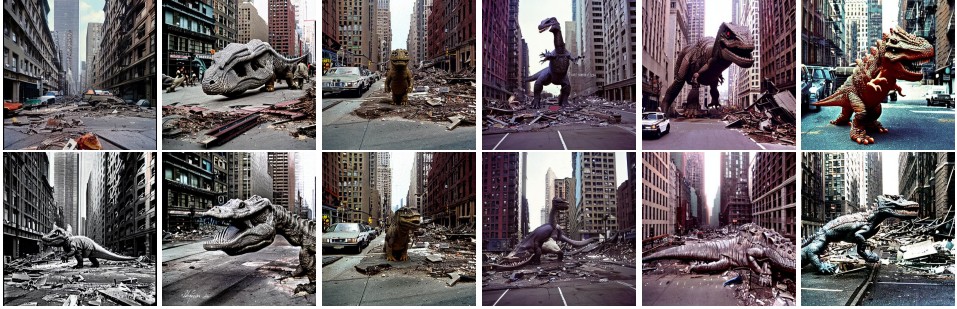

(b) Prompt: "dinosaur kaiju in nyc street, destroyed buildings, 1990s, photographic, kodak portra 400, 8k".

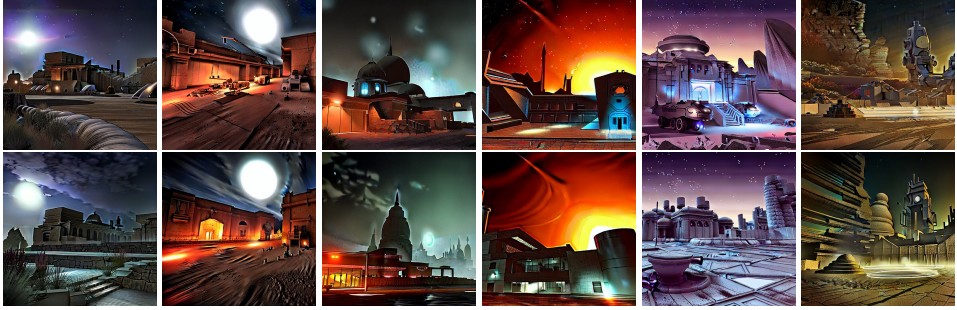

(c) Prompt: "a dark exterior landscape shot of jabba's palace at night, rusty mri machine star wars maschinen krieger, ilm, beeple, star citizen halo, mass effect, starship troopers, iron smelting pits, high tech industrial, warm saturated colours, dramatic space sky".

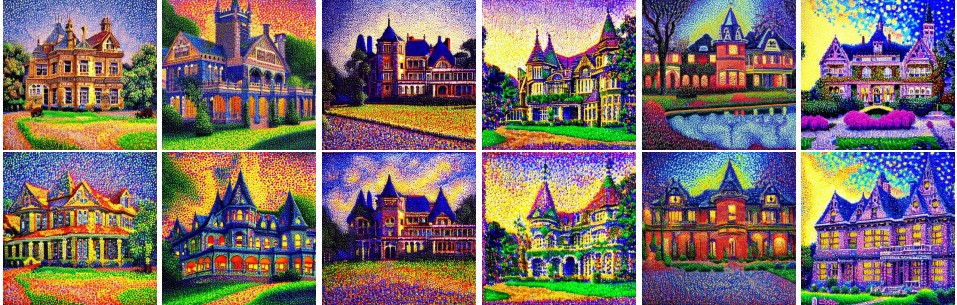

(d) Prompt: "Pointillism Painting of a Victorian manor at dusk soft glow HDR".

Figure A.8: Images generated using general prompts. For every subfigure, top row is generated using the original SD and bottom row is generated using the SD debiased for both gender and race. The pair of images at the same column are generated using the same noise.

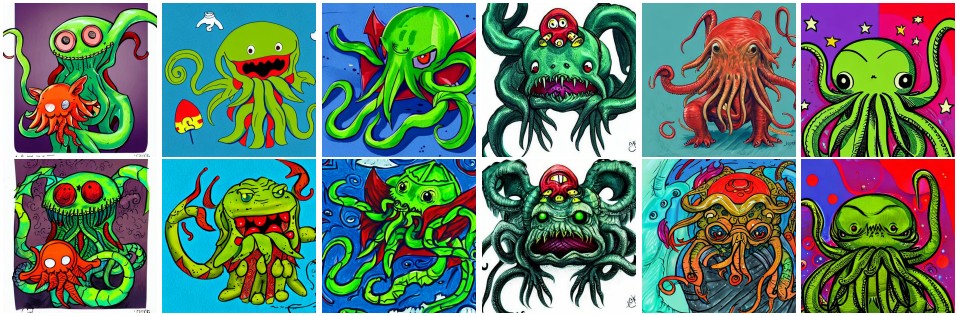

(a) Prompt: "a close up illustration of cthulhu as a cute kindergarten age monster playing with toys by artist jess bradley, concept art, digital art, gaudy colors ".

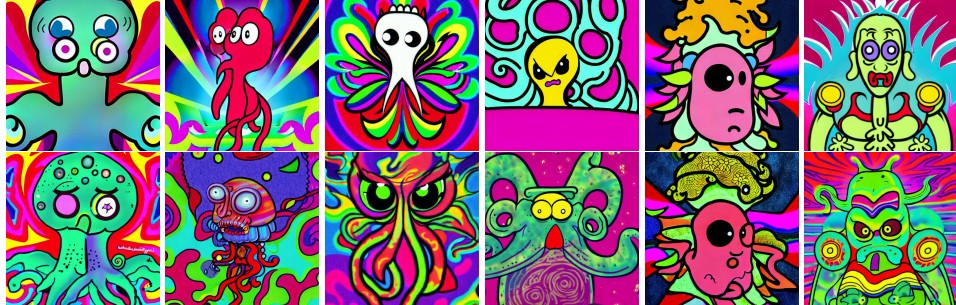

(b) Prompt: "deep focus shot of a crisp really angry squidward in the style of stephen gammel and lisa frank, dark psychedelica, psychedelic, anger, 8 k, award - winning art ".

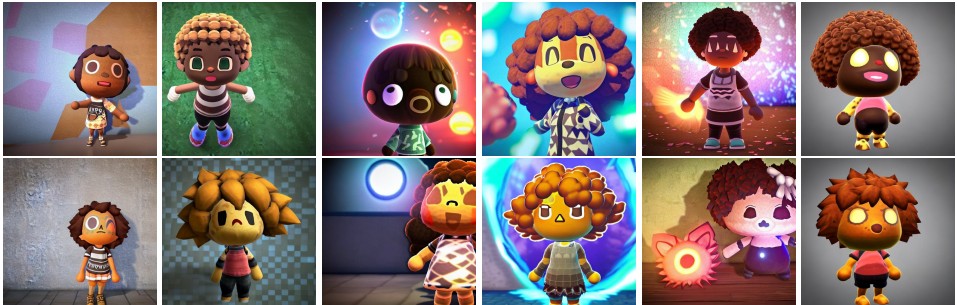

(c) Prompt: "3d octane render style glowing eyes 3d anime child model brown skin beautiful Afro hair 3d video game animal crossing background cinematic 8K".

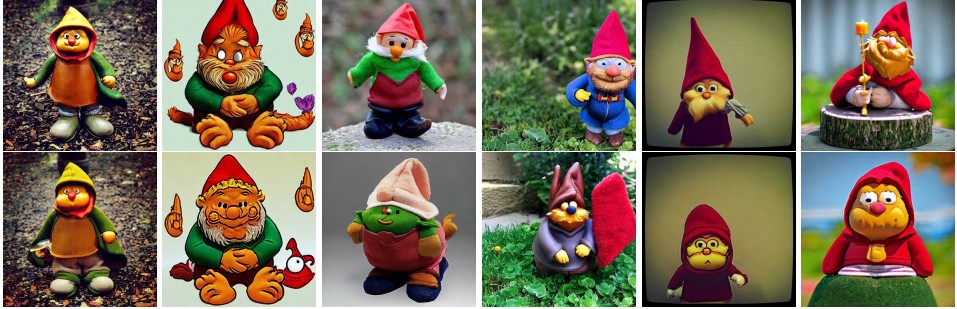

(d) Prompt: "\u201c garfield gnome \u201d ".

Figure A.9: Images generated using general prompts. For every subfigure, top row is generated using the original SD and bottom row is generated using the SD debiased for both gender and race. The pair of images at the same column are generated using the same noise.

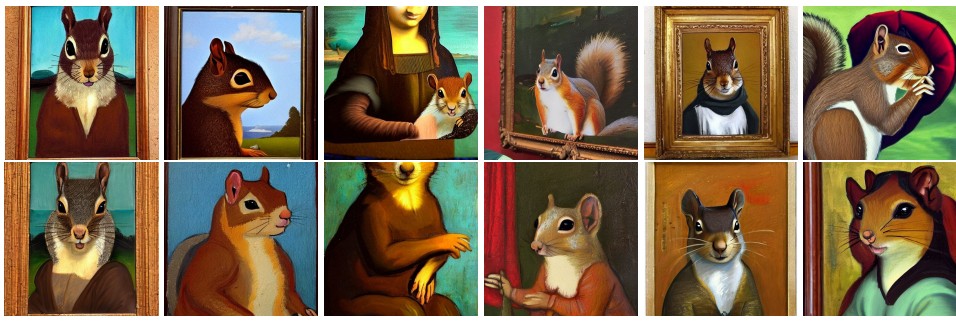

(a) Prompt: "Painting of a squirrel in the style of Mona Lisa".

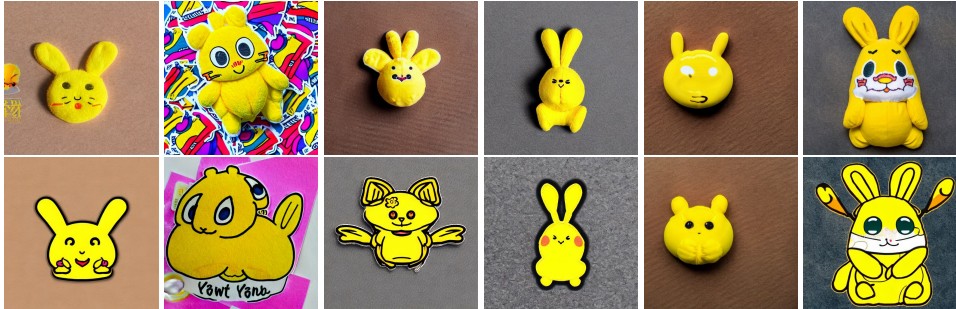

(b) Prompt: "macro shot top view cute yellow rabbit mascot with oversized eyes and ears, logo colored drawing sticker ".

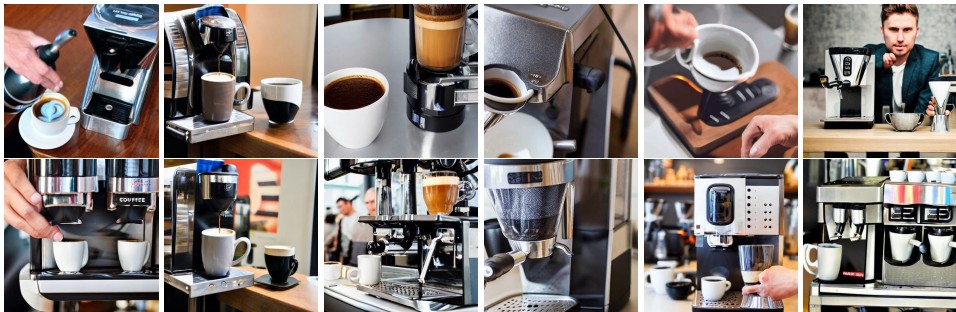

(c) Prompt: "a complex machine that pours coffee into your mouth when you wake up".

Figure A.10: Images generated using general prompts. For every subfigure, top row is generated using the original SD and bottom row is generated using the SD debiased for both gender and race. The image pairs at the same column are generated using the same noise.

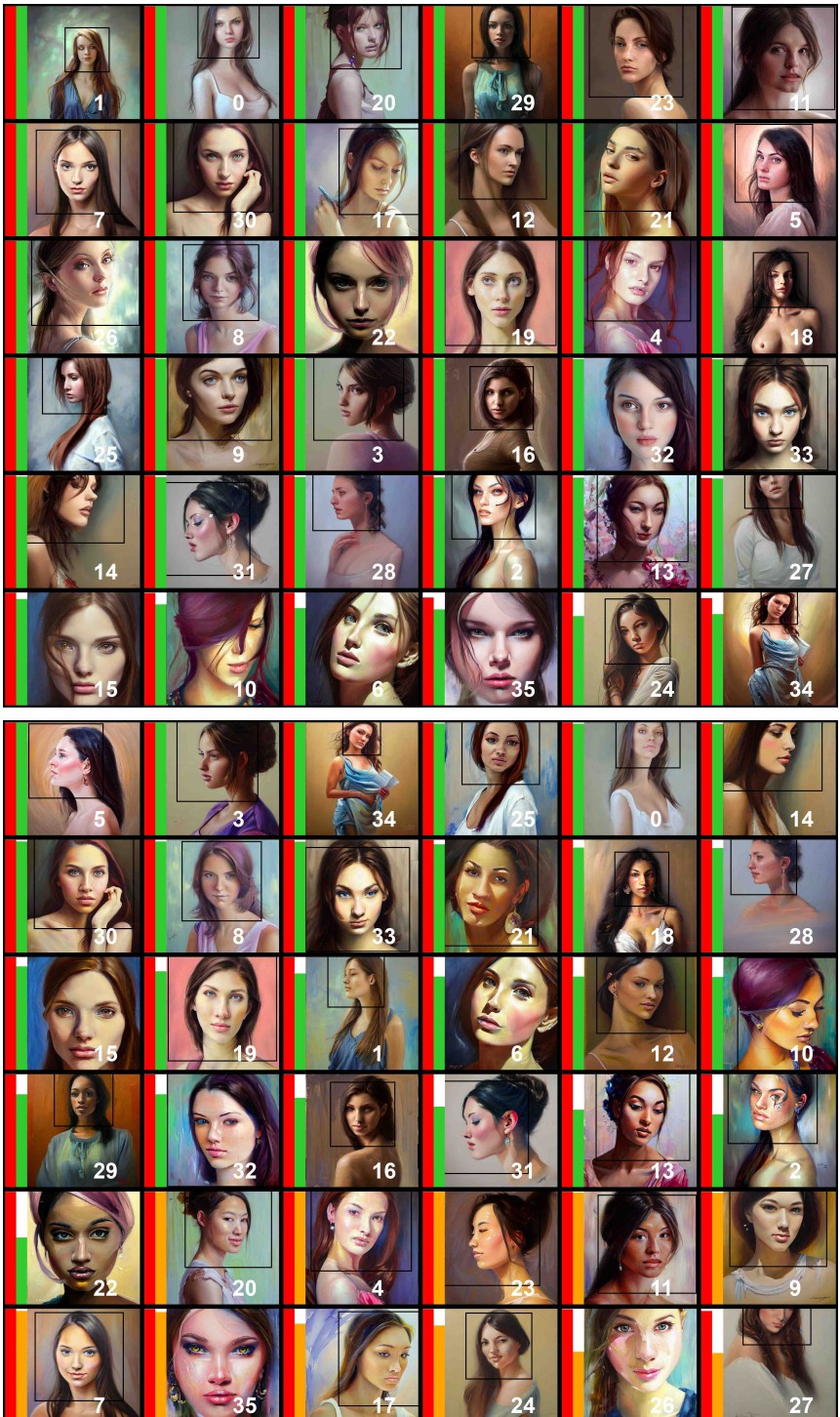

Figure A.11: Images generated from the prompt: "A beautiful painting of woman by Mandy Jurgens, Trending on artstation". Top/bottom 6×6 images are generated by the original/debiased SD. For every image, the first color-coded bar denotes the predicted gender: male or female. The second denotes race: WMELH, Asian, Black, or Indian. Bounding boxes denote detected faces. Bar height represents prediction confidence. The pair of images with the same number label are generated using the same noise.

```
pilot', 'jazz pianist', 'magician in the desert', 'model with
loose wavy curls', 'model with short hair and wavy side bangs',
'naval officer', 'painting of a nurse in a white uniform', 'paris
photographer', 'portrait of a flamenco dancer', 'scientist in his
lab working with chemicals', 'singapore wedding photographer',
'student with globe', 'the guitarist in Custom Picture Frame',
'top chef Seattle', 'wedding in venice photographer', 'wedding
photographer amsterdam', 'wedding photographer in Sydney',
'wedding photographer in tuscany'].
```

### A.7 EVALUATION OF DEBIASVL AND UCE

DebiasVL (Chuang et al., 2023) debiases vision-language models by projecting out biased directions in the text embeddings. Empirically, the authors apply it on Stable Diffusion v2-1[1] using the prompt template "A photo of a {occupation}.". They use 80 occupations for training and 20 for testing.

For the results reported in Table 1, we apply debiasVL on Stable Diffusion v1-5 using the prompt template "a photo of the of a {occupation}, a person". To debias gender or race individually, we use 1000 occupations for training and 50 for testing. To debias gender and race jointly, we use 500 occupations for training due to memory limit, and the same 50 occupations for testing. We use the same hyperparameter $\lambda = 500$ as in their paper.

We test this method for gender bias, with different diffusion models, training occupations, and prompt templates. Results are reported in Table 7. We find this method sensitive to both the diffusion model and the prompt. It generally works better for SD v2-1 than SD v1-5. Using a larger set of occupations for training might or might not be helpful. For some combinations, this method exacerbates rather than mitigates gender bias. We note that the failure of debiasVL is also observed in Kim et al. (2023).

For unified concept editing (UCE) (Gandikota et al., 2023) reported in Table 1, we use the same 37 occupations as from their paper and two templates, "{occupation}" and "a photo of a {occupation}", for training.

| Prompt Template | Model | Occupations | | Gender Bias ↓ |
| | | Train | Eval | |
|---|---|---|---|---|
| A photo of a {occupation}. | SD v2-1 | - | ours | 0.66±0.27 |
| | Debiased SD v2-1 | original | ours | 0.52±0.30 |
| | Debiased SD v2-1 | ours | ours | 0.78±0.21 |
| a photo of the face of a {occupation}, a person | SD v2-1 | - | ours | 0.67±0.31 |
| | Debiased SD v2-1 | original | ours | 0.49±0.28 |
| | Debiased SD v2-1 | ours | ours | 0.49±0.26 |
| A photo of a {occupation}. | SD v1-5 | - | ours | 0.61±0.26 |
| | Debiased SD v1-5 | original | ours | 0.92±0.12 |
| | Debiased SD v1-5 | ours | ours | 0.38±0.27 |
| a photo of the face of a {occupation}, a person | SD v1-5 | - | ours | 0.67±0.29 |
| | Debiased SD v1-5 | original | ours | 0.99±0.04 |
| | Debiased SD v1-5 | ours | ours | 0.98±0.10 |

Table 7: Evaluating debiasVL (Chuang et al., 2023) with different diffusion models, training occupations, and prompt templates.

---

[1] https://huggingface.co/stabilityai/stable-diffusion-2

## A.8 EXPANDED VERSION OF FIG 3 FROM MAIN TEXT

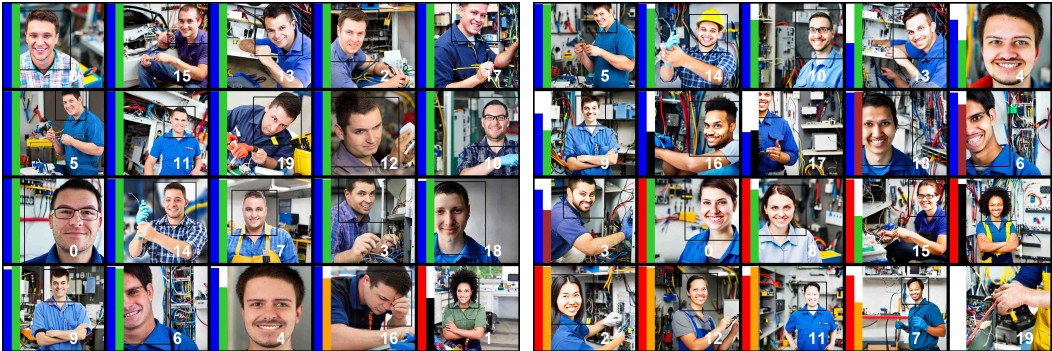

(a) Prompt w/ unseen occupation: "a photo of the face of a electrical and electronics repairer, a person". Gender bias: 0.84 (original) → 0.11 (debiased). Racial bias: 0.48 → 0.10. Gender×Race bias: 0.24 → 0.06.

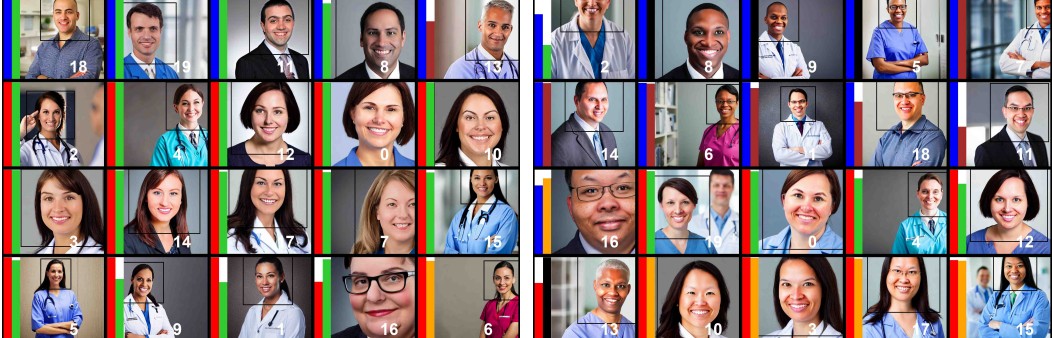

(b) Prompt w/ unseen occupation: "a photo of the face of a cardiologist, a person". Gender bias: 0 (original) → 0.10 (debiased). Racial bias: 0.44 → 0.08. Gender×Race bias: 0.19 → 0.06.

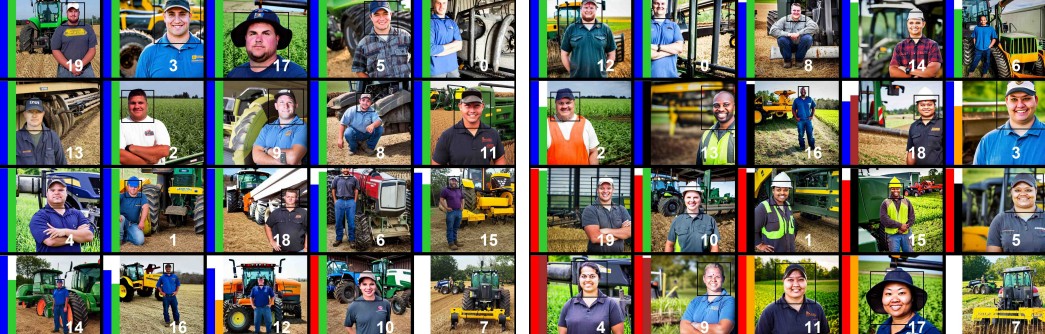

(c) Prompt w/ unseen occupation: "a photo of the face of a farm equipment service technician, a person". Gender bias: 0.95 (original) → 0.10 (debiased). Racial bias: 0.48 → 0.11. Gender×Race bias: 0.24→ 0.06.

Figure A.12: Images generated from the original model (left) and the model jointly debiased for gender and race (right). The model is debiased using the prompt template "a photo of the face of a {occupation}, a person". For every image, the first color-coded bar denotes the predicted gender: male or female. The second denotes race: WMELH, Asian, Black, or Indian. Bounding boxes denote detected faces. Bar height represents prediction confidence. For the same prompt, images with the same number label are generated using the same noise.

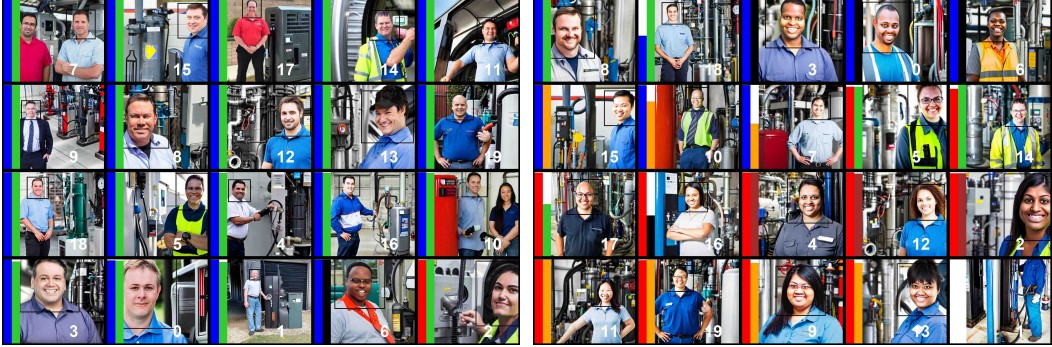

(a) Prompt w/ unseen occupation: "a photo of the face of a gas compressor and gas pumping station operator, a person". Gender bias: 0.81 (original) → 0.14 (debiased). Racial bias: 0.46 → 0.06. Gender×Race bias: 0.23 → 0.05.

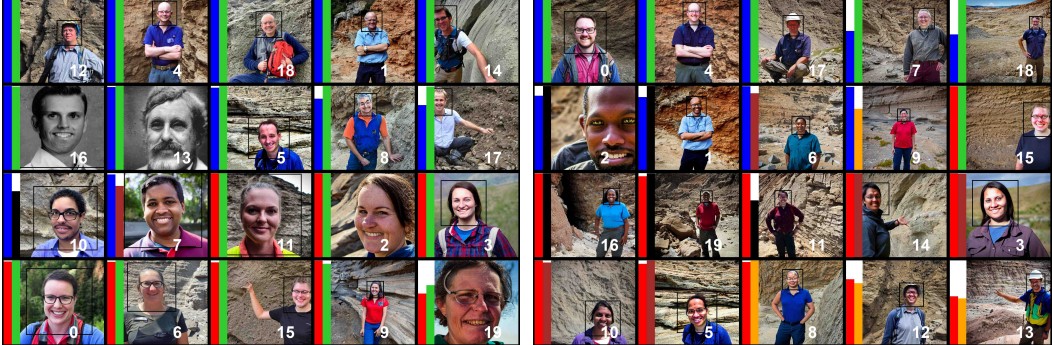

(b) Prompt w/ unseen occupation: "a photo of the face of a geologist, a person". Gender bias: 0.23 (original) → 0.01 (debiased). Racial bias: 0.47 → 0.11. Gender×Race bias: 0.21 → 0.06.

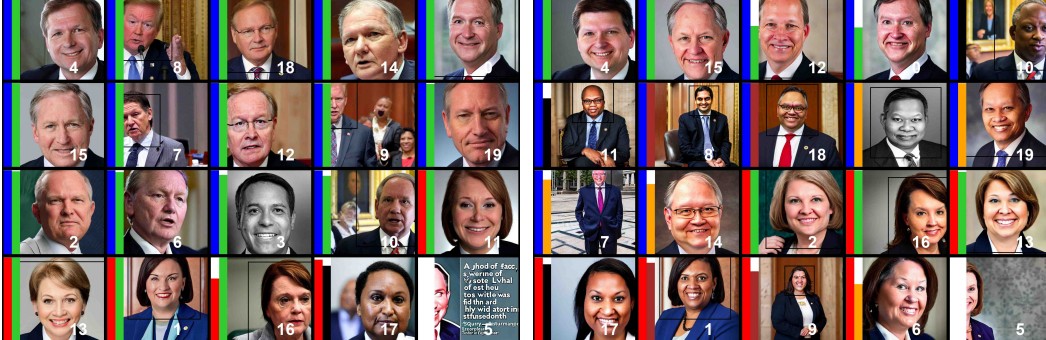

(c) Prompt w/ unseen occupation: "a photo of the face of a senator, a person". Gender bias: 0.87 (original) → 0.17 (debiased). Racial bias: 0.49 → 0.33. Gender×Race bias: 0.24→ 0.15.

Figure A.13: Images generated from the original model (left) and the model jointly debiased for gender and race (right). The model is debiased using the prompt template "a photo of the face of a {occupation}, a person". For every image, the first color-coded bar denotes the predicted gender: male or female. The second denotes race: WMELH, Asian, Black, or Indian. Bounding boxes denote detected faces. Bar height represents prediction confidence. For the same prompt, images with the same number label are generated using the same noise.

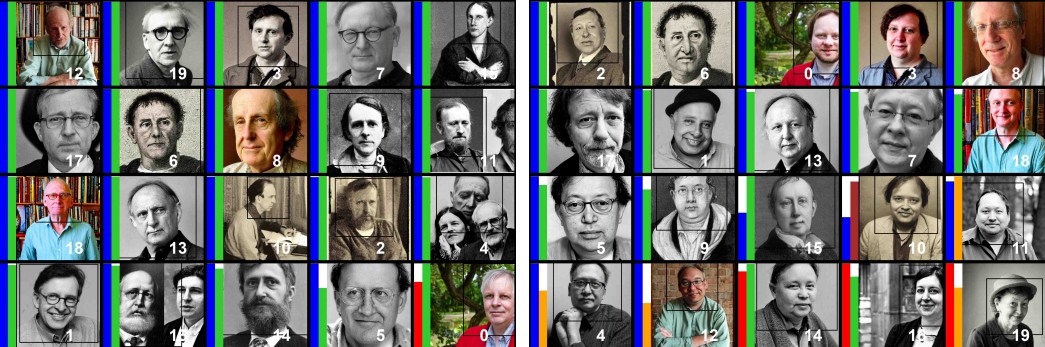

(a) Prompt w/ unseen style or context: "English writer and essayist". Gender bias: 0.86 (original) → 0.37 (debiased). Racial bias: 0.50 → 0.38. Gender×Race bias: 0.24 → 0.18.

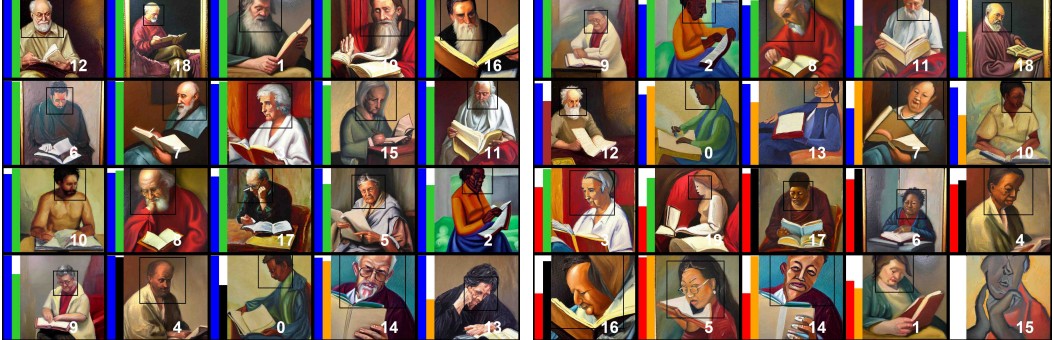

(b) Prompt w/ unseen style or context: "A philosopher reading. Oil painting.". Gender bias: 0.80 (original) → 0.23 (debiased). Racial bias: 0.45 → 0.31. Gender×Race bias: 0.22 → 0.15.

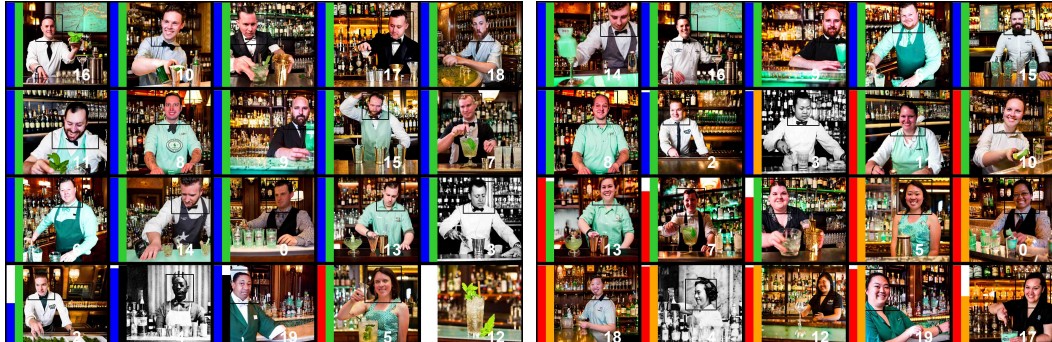

(c) Prompt w/ unseen style or context: "bartender at willard intercontinental makes mint julep". Gender bias: 0.87 (original) → 0.17 (debiased). Racial bias: 0.49 → 0.33. Gender×Race bias: 0.24→ 0.15.

Figure A.14: Images generated from the original model (left) and the model jointly debiased for gender and race (right). The model is debiased using the prompt template "a photo of the face of a {occupation}, a person". For every image, the first color-coded bar denotes the predicted gender: male or female. The second denotes race: WMELH, Asian, Black, or Indian. Bounding boxes denote detected faces. Bar height represents prediction confidence. For the same prompt, images with the same number label are generated using the same noise.

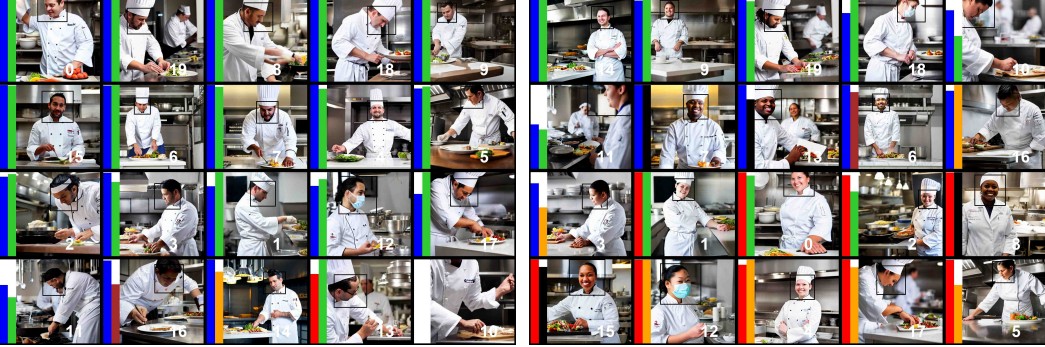

(a) Prompt w/ unseen occupation: "A chef in a white coat leans on a table". Gender bias: 0.84 (original) → 0.03 (debiased). Racial bias: 0.45 → 0.32. Gender×Race bias: 0.22 → 0.14.

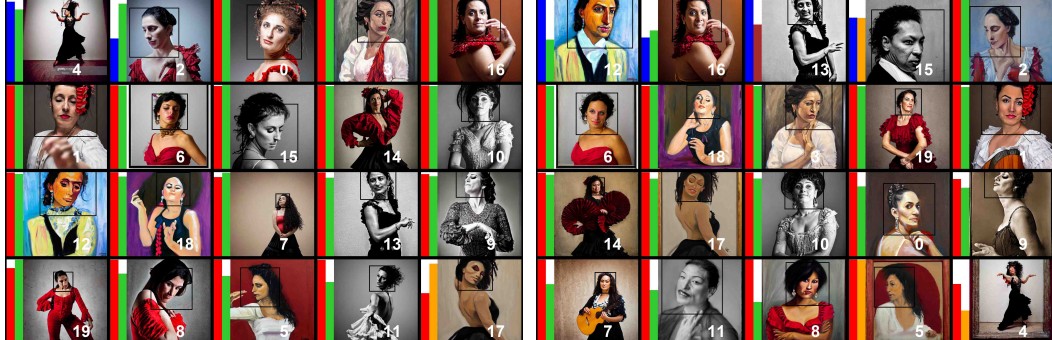

(b) Prompt w/ unseen style: "portrait of a flamenco dancer". Gender bias: 0.76 (original) → 0.59 (debiased). Racial bias: 0.47 → 0.38. Gender×Race bias: 0.23 → 0.18.

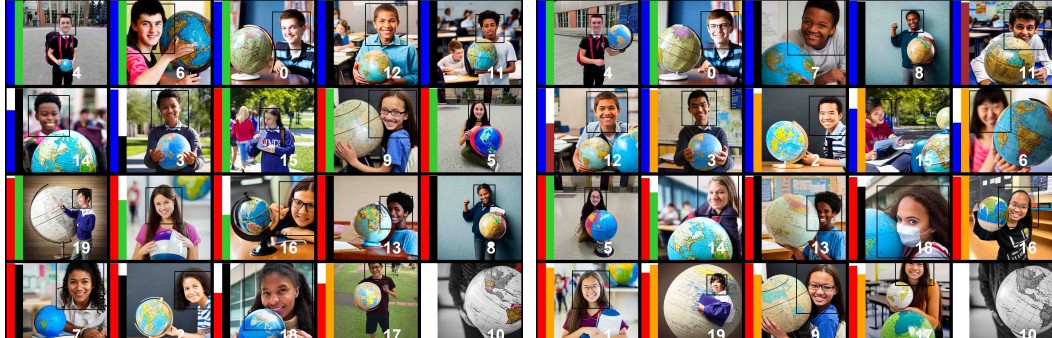

(c) Prompt w/ unseen context: "student with globe". Gender bias: 0.23 (original) → 0.04 (debiased). Racial bias: 0.33 → 0.21. Gender×Race bias: 0.15→ 0.09.

Figure A.15: Images generated from the original model (left) and the model jointly debiased for gender and race (right). The model is debiased using the prompt template "a photo of the face of a {occupation}, a person". For every image, the first color-coded bar denotes the predicted gender: male or female. The second denotes race: WMELH, Asian, Black, or Indian. Bounding boxes denote detected faces. Bar height represents prediction confidence. For the same prompt, images with the same number label are generated using the same noise.

## A.9 MULTI-FACE IMAGE GENERATIONS

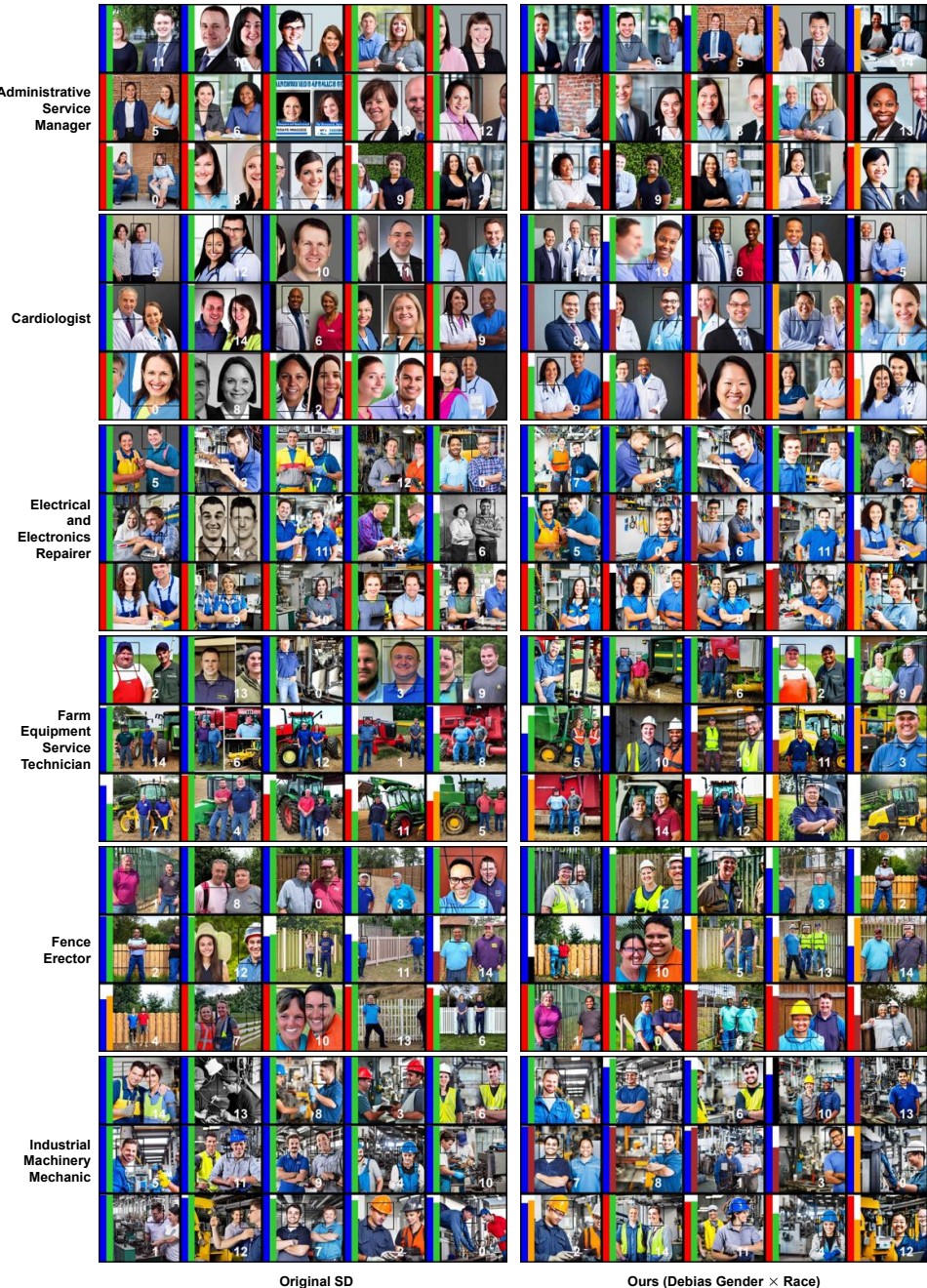

Figure A.16: Examples of image generation featuring two individuals. Images are generated using the prompt "A photo of the faces of two {occupation}, two people". The occupation is shown at the left. For every occupation, images with the same number are generated using the same noise. For every image, the first color-coded bar denotes the predicted gender: male or female. The second denotes race: WMELH, Asian, Black, or Indian. Bar height represents prediction confidence. Bounding boxes denote the faces that covers the largest area in every image.

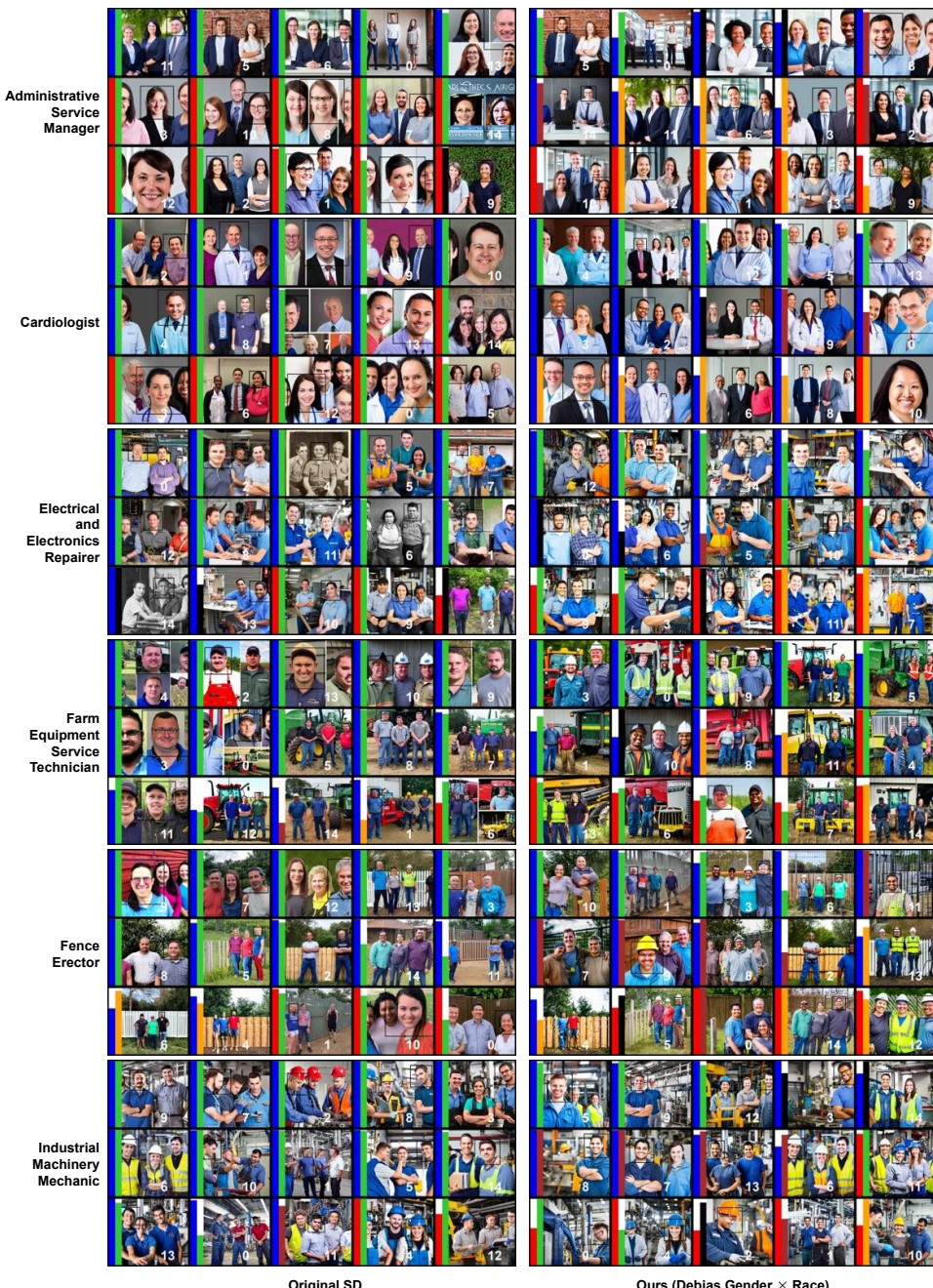

Figure A.17: Examples of image generation featuring three individuals. Images are generated using the prompt "A photo of the faces of three {occupation}, three people". The occupation is shown at the left. For every occupation, images with the same number are generated using the same noise. For every image, the first color-coded bar denotes the predicted gender: male or female. The second denotes race: WMELH, Asian, Black, or Indian. Bar height represents prediction confidence. Bounding boxes denote the faces that covers the largest area in every image.

A.10 COMPARING DIFFERENT FINETUNED COMPONENTS

Fig. A.18, A.19, A.20 compare generated images from finetuning different components of the diffusion model.

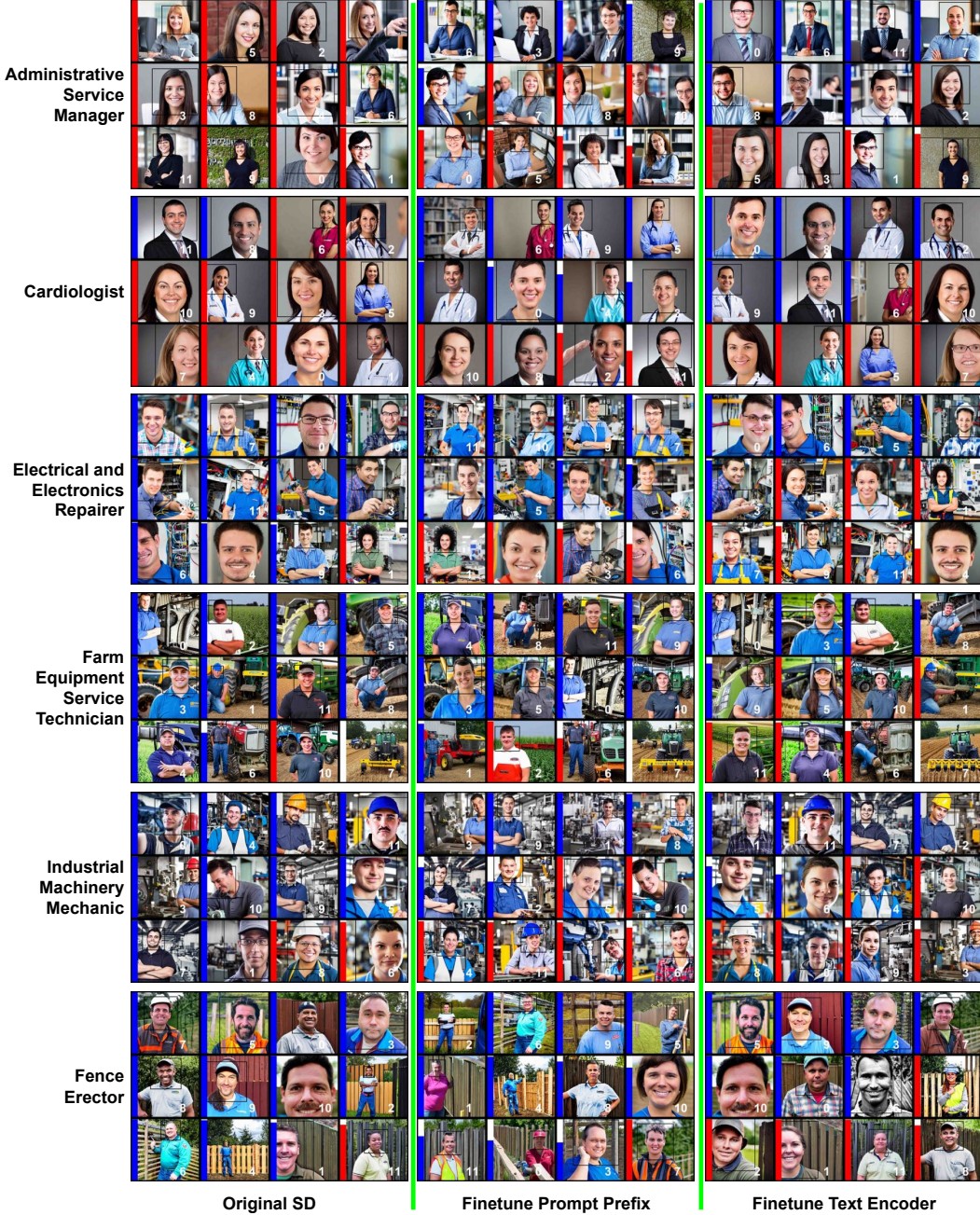

Figure A.18: Examples of generated images when different components of diffusion model are finetuned. Images are generated using the prompt "a photo of the face of a {occupation}, a person". The occupation is shown at the left. For every occupation, images with the same number are generated using the same noise.

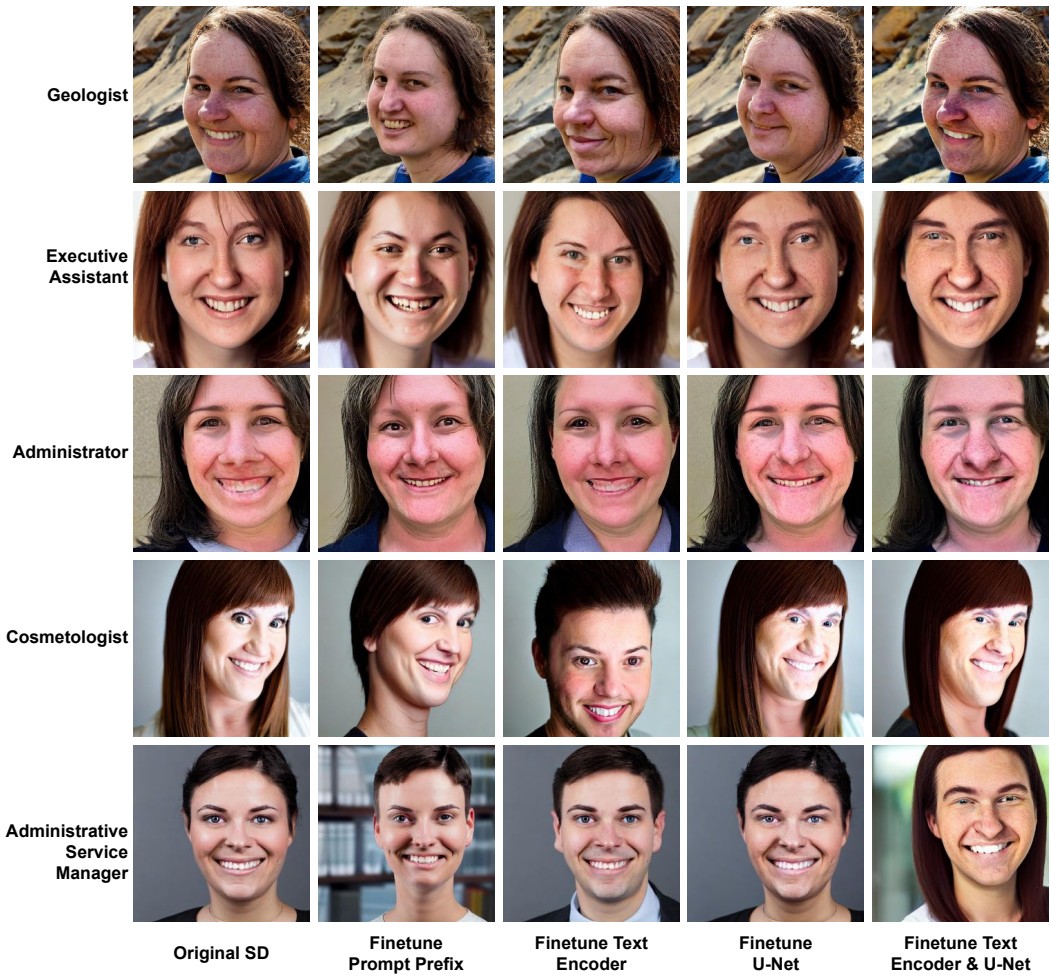

Figure A.19: Examples showing how finetuning U-Net may deteriorate image quality regarding facial skin texture. Images are generated using the prompt "`a photo of the face of a {occupation}, a person`". The occupation is shown at the left side of every row. Every row of images are generated using the same noise.

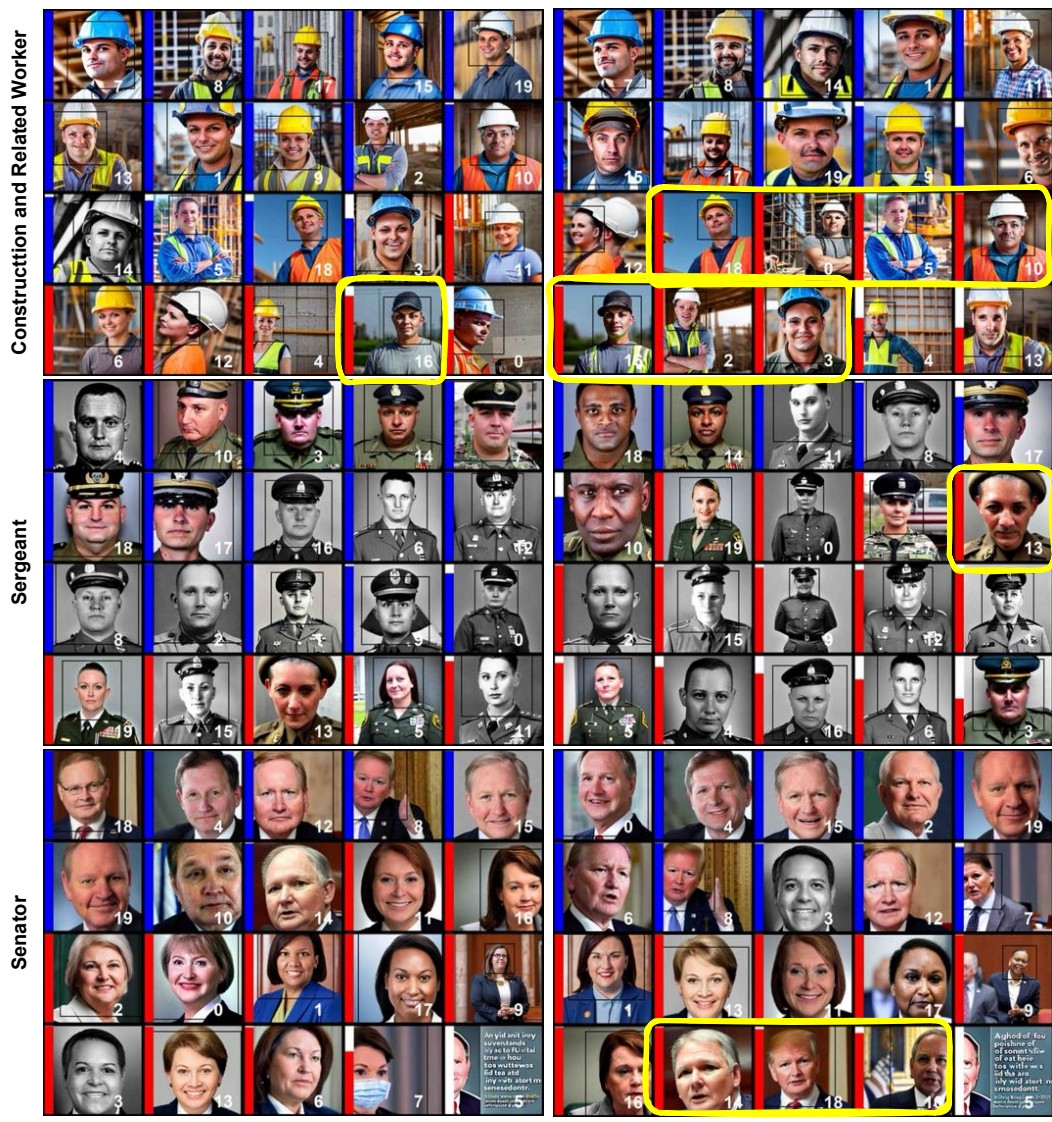

Figure A.20: Examples showing how finetuning U-Net may generate images whose predicted gender according to the classifier does not agree with human perception, *i.e.*, overfitting. Each image is accompanied by a color-coded bar on the left to indicate the predicted gender: blue for male and red for female. The height of the bar represents the classifier's prediction confidence. The lead author reviewed these generated images. Those where the predicted gender didn't match their perception are highlighted in yellow boxes. Images are generated using the prompt "`a photo of the face of a {occupation}, a person`". The occupation is shown at the left. For every occupation, images with the same number are generated using the same noise.

### A.11 AGE ALIGNMENT

For the experiment that aligns age distribution while simultaneously debiasing gender and race, we train a classifier that classifies gender, race, and age using the FairFace dataset. We use FairFace faces as external faces for the face realism preserving loss. We set $\lambda_{face} = 0.1$ and $\lambda_{img,1} = 8$. For the gender attribute, we use $\lambda_{img,2} = 0.2 \times \lambda_{img,1}$, and $\lambda_{img,3} = 0.2 \times \lambda_{img,2}$. For race and age, we use $\lambda_{img,2} = 0.3 \times \lambda_{img,1}$, and $\lambda_{img,3} = 0.3 \times \lambda_{img,2}$. We use batch size $N = 40$ and set the confidence threshold for the distributional alignment loss $C = 0.7$. We train for 14k iterations using AdamW optimizer with learning rate 5e-5. We checkpoint every 200 iterations and report the best checkpoint. The finetuning takes around 56 hours on 8 NVIDIA A100 GPUs.

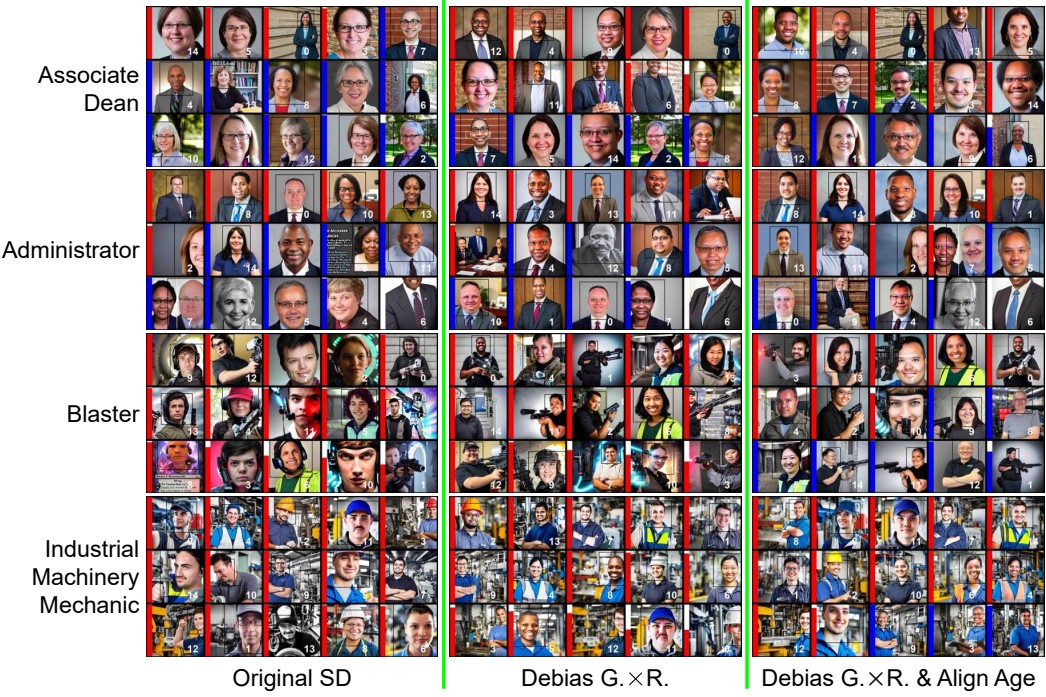

Figure A.21: Examples showing the effect of aligning age to 75% young and 25% old besides jointly debiasing gender and race. In this figure, the color-coded bar denotes age: red is yound and blue is old. We do not annotate gender and race for visual clarity. Images are generated using the prompt "a photo of the face of a {occupation}, a person". The occupation is shown at the left. For every occupation, images with the same number are generated using the same noise.

### A.12 DEBIASING MULTIPLE CONCEPTS AT ONCE

Below we list the prompts used for training and testing.

(1) Occupational prompts: formulated with the template "a photo of the face of a {occupation}, a person". We utilize the same 1000/50 occupations as in Section 5.1 for training/testing. The occupations are listed in Sec. A.2.

(2) Sports prompts: formulated with the template "a person playing {sport}". We use 250/50 sports activities for training/testing. Training sports include: ['ulama', 'casting (fishing)', 'futsal', 'freestyle slalom skating', 'figure skating', 'dinghy sailing', 'skipping rope', 'kickboxing', 'cross-country equestrianism', 'limited overs cricket', 'eskrima', 'equestrian vaulting', 'creeking', 'sledding', 'capoeira', 'enduro', 'ringette', 'bodyboarding', 'sumo', 'valencian pilota', 'hunting', 'jetsprint', 'fives', 'laser tag', ···]. Test sports are ['pommel horse', 'riverboarding', 'hurdles', 'underwater

hockey', 'broomball', 'running', 'vovinam', 'rock fishing', 'barrel racing', 'cross-country cycling', 'silat', 'canoeing', 'cowboy action shooting', 'telemark skiing', 'adventure racing', 'olympic weightlifting', 'wiffle ball', 'toboggan', 'rhythmic gymnastics', 'english pleasure', 'northern praying mantis (martial art)', 'aggressive inline skating', 'arena football', 'australian rules football', 'beach tennis', 'haidong gumdo', 'trial', 'bandy', 'ball (rhythmic gymnastics)', 'bujinkan', 'freestyle football', 'gaelic football', 'horseball', 'okinawan kobudō', 'slamball', 'pankration', 'fox hunting', 'street football', 'juggling club', 'land sailing', 'ultimate (sport)', 'skibobbing', 'test cricket', 'bikejoring', 'tang soo do', 'sambo (martial art)', 'wing chun', 'synchronized swimming', 'rink bandy', 'beach handball', 'cyclo-cross', 'harness racing', 'jujutsu', 'slacklining', 'polo', 'rugby', 'association football', 'medley swimming', 'big-game fishing', 'demolition derby', 'rope (rhythmic gymnastics)', 'taekwondo', 'team handball', 'cross-country skiing', 'rundown', 'schutzhund', 'canoe polo', 'archery', 'squash (sport)', 'snooker', 'wing tsun', 'jai alai', 'streetball', 'sea kayak', 'muay thai', 'lure coursing', 'calisthenics', 'krav maga', 'wheelchair basketball', 'trampolining', 'indoor american football', 'speed skating', 'amateur wrestling', 'rugby sevens', 'frontenis'].

(3) Occupational prompts with style or context: these are non-templated prompts that specify occupations with diverse styles or contexts. We train/test on 150/19 such prompts obtained from the captions in the LAION-AESTHETICS dataset. The training prompts include: ["a epic hero adventurer holding a torch in a dark cave, artgerm, realistic, cryengine, symmetric", "salvador dali the painter became super saiyan, dragon ball style, cinematic lighting, highly detailed, concept art, art by wlop and artgerm and greg rutkowski, masterpiece, trending on artstation, 8 k", "concept art of scientist with scifi accessories by jama jurabaev, brush stroke,, trending on artstation, upper half portrait, symmetry, high quality, extremely detailed", "detective falling through the sky, city, by peter mohrbacher, artgerm, karol bak, loish, ayami kojima, james stokoe, highly detailed, ultra detailed, ultra realistic, trending on artstation", "concept art of agent 4 7, vector art, by cristiano siqueira, brush hard, highly detailed, artstation, high quality", "nightbringer yasuo slashing, ultra details background trending on artstation digital painting splashart drawn by a professional artist", "portrait of a middle - aged writer with a beard, he is smoking a cigarette, style of greg rutkowski", "cute star trek officer lady gaga, natural lighting, path traced, ···]. The test prompts are: ["concept art of elite scientist by jama jurabaev, emperor secret society, cinematic shot, trending on artstation, high quality, brush stroke", "cyborg scientist by jama jurabaev, cinematic shot, extremely detailed, trending on artstation, high quality, brush stroke", "a haggard detective in a trenchcoat scanning a crimescene, sketchy artstyle, digital art, dramatic, thick lines, rough lines, line art, cinematic, trending on artstation", "computer scientist who served as an intel systems engineer, full-body shot, digital painting, smooth, elegant, hd, art by WLOP and Artgerm and Greg Rutkowski and Alphonse Mucha", "a painting so beautiful and universally loved it creates peace on earth, profound epiphany, trending on artstation, by john singer sargent", "a portrait of fish magician in glass armor releasing spell, full height, moving forward, concept art, trending on artstation, highly detailed,

intricate, sharp focus, digital art, 8 k", "blonde sailor moon
as aeon flux, by Stanley Artgerm Lau, greg rutkowski, Craig
mullins, Peter chung, thomas kindkade, alphonse mucha, loish,",
"a aesthetic portrait of a magician working on ancient machines
to do magic, concept art", "portrait old barbarian warrior with
trucker mustache and short hair, 8 k, trending on art station,
by tooth wu and greg rutkowski", "High fantasy detective with
whips with crab companion, RPG Scene, Oil Painting, octane render,
Trending on Artstation, Insanely Detailed, 8k, UHD", "selfie
of a space soldier by louis daguerre, cinematic, high quality,
cgsociety, artgerm, 4 k, uhd, 5 0 mm, trending on artstation",
"a beautiful model in crop top, by guweiz and wlop and ilya
kuvshinov and artgerm, symmetrical eyes, aesthetic, gorgeous,
stunning, alluring, attractive, artstation, deviantart, pinterest,
digital art", "a mad scientist mutating into a monster because of
spilled chemicals in the laboratory, wlop, trending on artstation,
deviantart, anime key visual, official media, professional art,
8 k uhd", "portrait of a mutant wrestler with posing in front of
muscle truck, with a spray painted makrel on it, dystopic, dust,
intricate, highly detailed, concept art, Octane render", "portrait
of a vicotrian doctor in suit with helmet by darek zabrocki and
greg ruthkowski, alphonse mucha, simon stalenhag and cinematic and
atmospheric, concept art, artstation, trending on artstation",
"concept art of portrait ofcyborg scientist by jama jurabaev,
extremely detailed, trending on artstation, high quality, brush
stroke", "a beautiful masterpiece painting of a clothed artist by
juan gimenez, award winning, trending on artstation,", "comic book
boss fight, highly detailed, professional digital painting, Unreal
Engine 5, Photorealism, HD quality, 8k resolution, cinema 4d, 3D,
cinematic, art by artgerm and greg rutkowski", "magician shuffling
cards, cards, fantasy, digital art, soft lighting, concept art, 8
k"].

(4) personal descriptors: these prompts describe individual(s). We use 40/10 such prompts for
training/testing. The training prompts are: ["Business person looking at wall with
light tunnel opening", "person sitting on rock on body of water",
"person standing on rocky cliff", "Oil painting of a person on
a horse", "Cleaning service person avatar cartoon character", "A
person sits against a wall in Wuhan, China.", "person riding on
teal dutch bicycle", "Most interesting person", "person standing
beside another person holding fire poi", "Youngest person reach
South Pole", "A mural of a person holding a camera.", "painting
of two dancing persons", "elderly personal care maryland", "person
doing fire dancing", "three persons standing near the edge of a
cliff during day", "person throwing fish net while standing on
boat", "person in black and white shirt lying on yellow bed",
"A person playing music.", "person dances in door way with a
view of the taj mahal", "person drinking out of a lake with
lifestraw", "painting of a person standing outside a cottage",
"person standing on mountaintop arms spread", "a person standing
in front of a large city landscape", "person standing in front
of waterfall during daytime", "person lying on red hammock",
"Front view of person on railroad track between valley", "A
person flying through the air on a rock", "A person cycling to
work through the city", "person with colorful balloons", "person
decorating a wedding cake", "person standing in front of Torii
Gate", "person wearing the headphones on the street", "person
sitting on a rock", "Colourful stage set with person in costume",
"person standing beside trees during winter season", "person on

```
a mountain top", "An image of an elderly person painting", "Day
6:  An old person", "iluminated tent with a person sitting out
front", "person sitting alone at a street stall eating soup"].
```
The test prompts are: `["bird's eyeview photo of person lying on green`
`grass", "A person holding a picture in front of a desert.", "A`
`painting of a person in a garage.", "steel wool photography of`
`person in room", "individual photo shoot in Prague", "Oil painting`
`of a person wearing colorful fabric", "person standing in front`
`of cave", "person in cold weather in a tent", "A person sitting`
`on dry barren dirt.", "a person standing next to a vase of flowers`
`on a table", "hot personal trainer", "a person lying on a dog",`
`"Image may contain:  person, flower and sunflower", "person in`
`water throwing guitar", "person standing at a forge holding a`
`sledge hammer", "image of a homeless person sitting on the side`
`of a building", "H&M spokesperson:  'Our models are too thin'",`
`"Biohazard cleaning persons", "A close up of a person wearing a`
`hat", "photo of person covered by red headscarf"].`

## A.13 GENERATED IMAGES FROM DEBIASING MULTIPLE CONCEPTS

In Figure A.22, we first compare the generated images for occupational prompts by the original SD, the SD debiased for single concept, and the SD debiased for multiple concepts. We find that multi-concept debiasing seems to increase the likelihood of generating images that blend male and female characteristics. Notable examples include: Salesperson images No. 6, 12, and 15, which appear male but have hairstyles typically associated with females; violinist images No. 1, 8, and 2, perceived as male but dressed in clothing typically associated with females; and community and social service specialist image No. 10, which is perceptually male but with pink lipstick, a feature commonly associated with females. We did not observe other significant impacts on image quality resulting from the debiasing of multiple concepts.

We show generated images for occupational prompts in Figure A.23, for sports prompts in Figure A.24, for occupational prompts with style or context in Figure A.25, and for personal descriptors in Figure A.26.

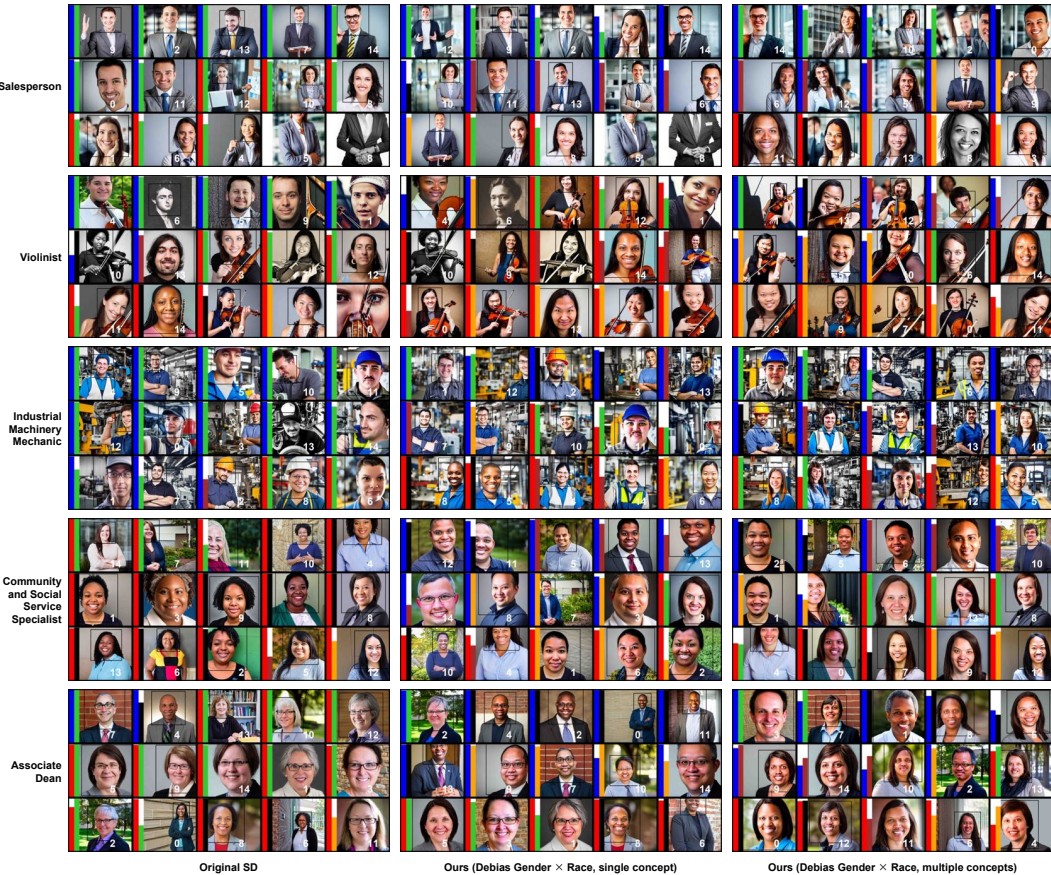

Figure A.22: Comparison of generated images for occupational prompts between original SD, the SD debiased for single concept, and the SD debiased for multiple concepts. Images are generated using the prompt template "a photo of the face of a {occupation}, a person". The occupation is shown at the left. For every occupation, images with the same number are generated using the same noise.

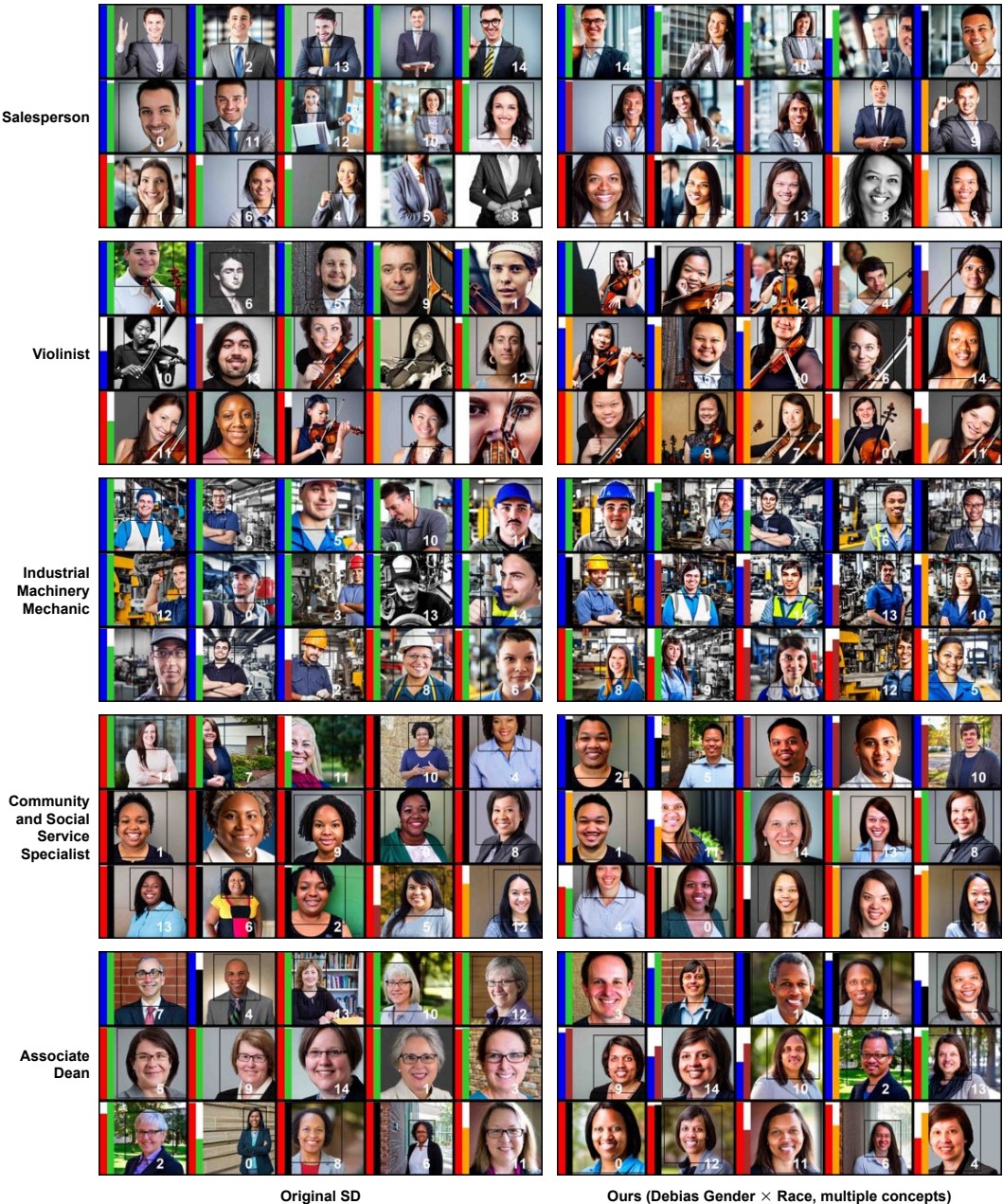

Figure A.23: Comparison of images generated for occupational prompts by original SD and the SD debiased for multiple concepts. Images are generated using the prompt template "a photo of the face of a {occupation}, a person". The occupation is shown at the left. For every prompt, images with the same number are generated using the same noise.

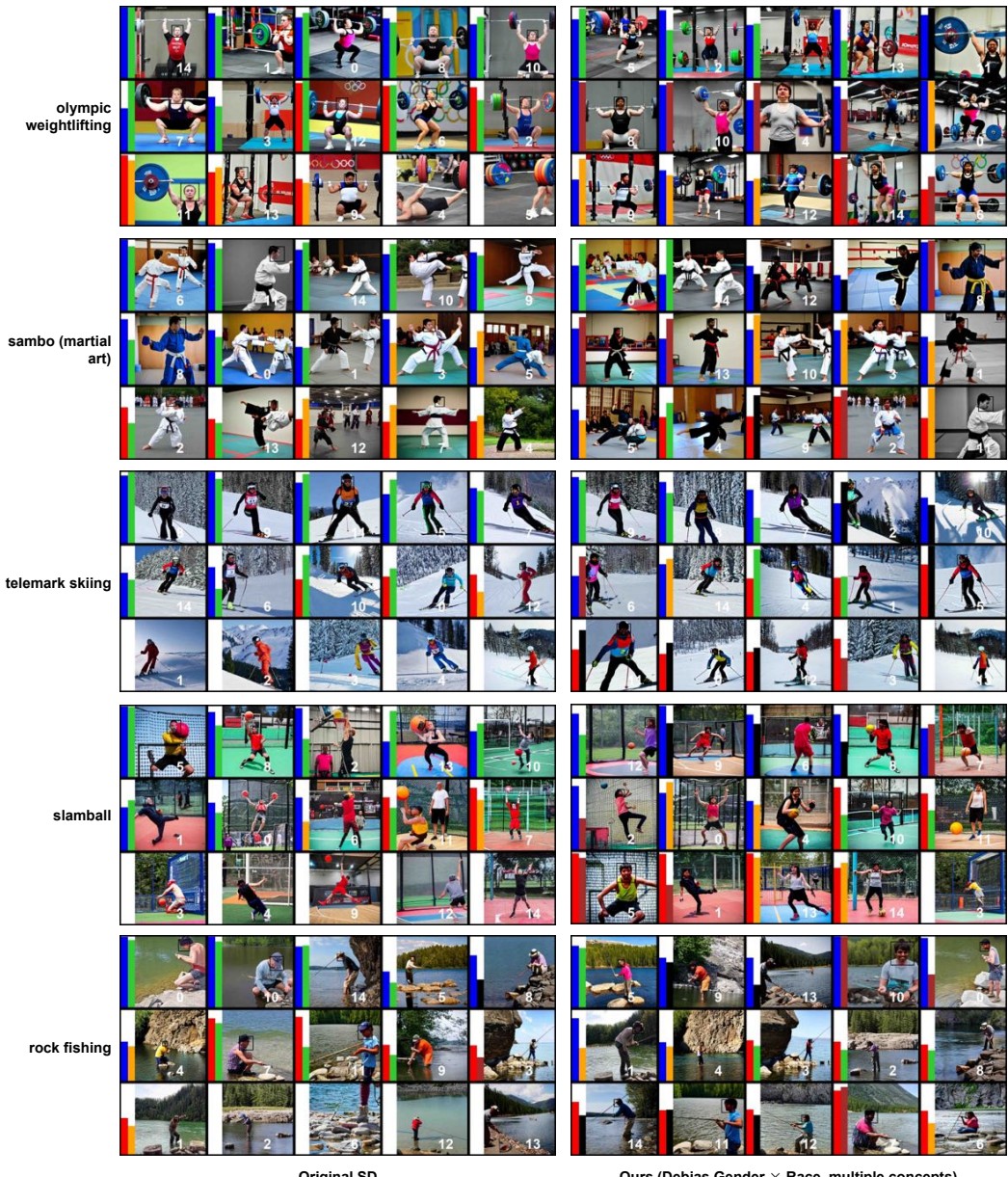

Figure A.24: Comparison of images generated for sports prompts by original SD and the SD debiased for multiple concepts. Images are generated using the prompt template "a person playing {sport}". The sport is shown at the left. For every prompt, images with the same number are generated using the same noise.

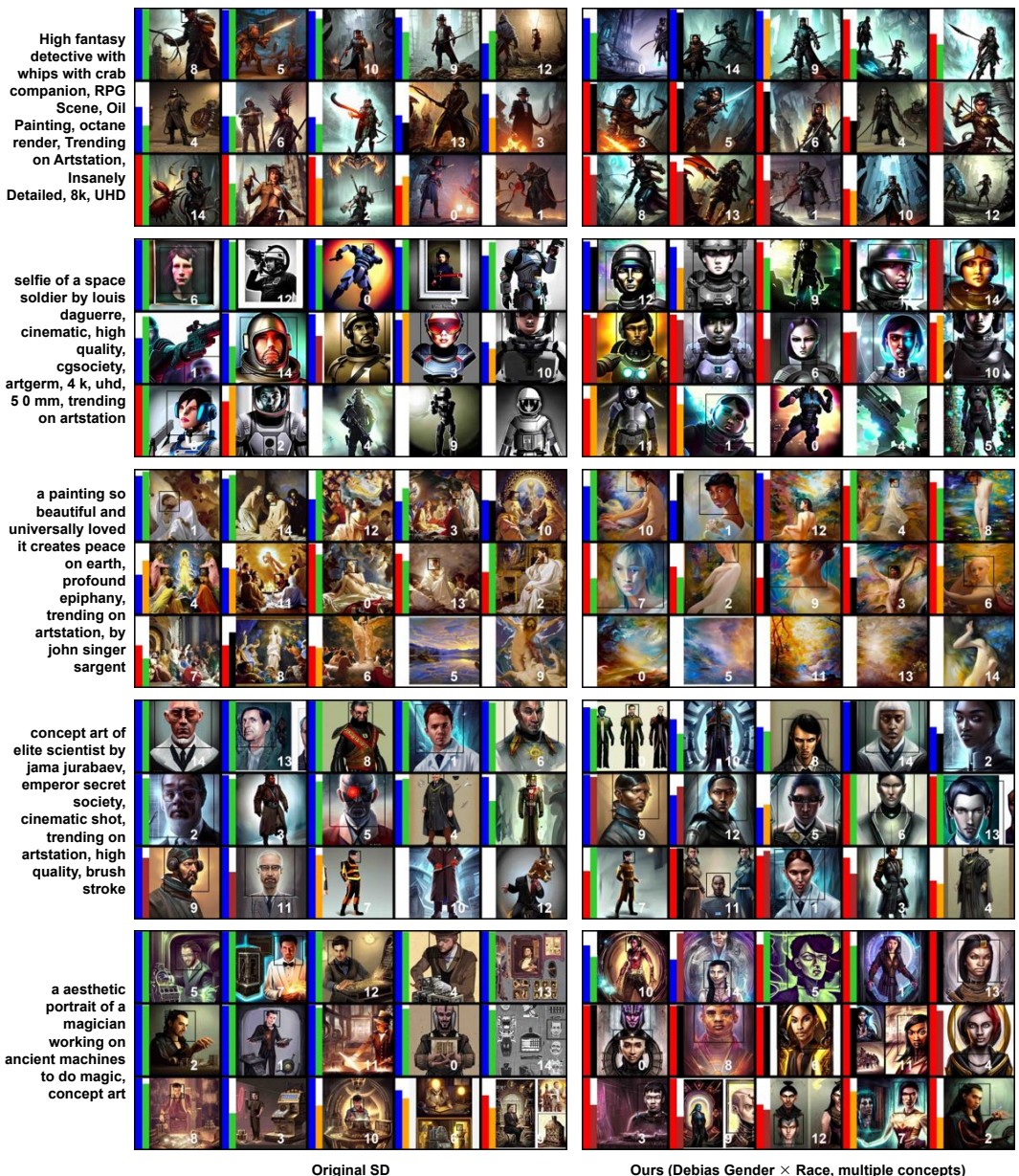

Figure A.25: Comparison of images generated for occupational prompts with style or context by original SD and the SD debiased for multiple concepts. The prompts are shown at the left. For every prompt, images with the same number are generated using the same noise.

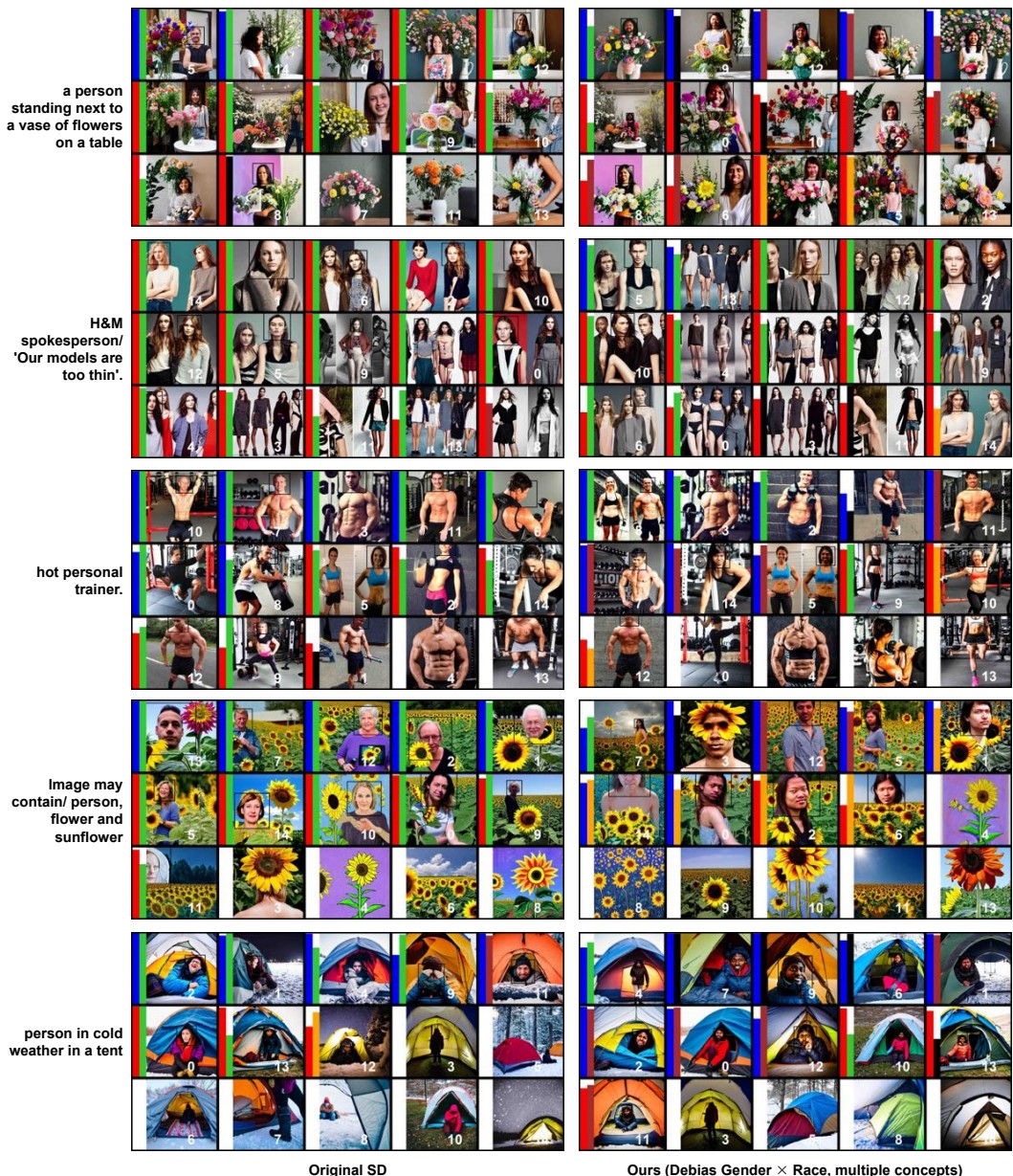

Figure A.26: Comparison of images generated for personal descriptors by original SD and the SD debiased for multiple concepts. The prompts are shown at the left. For every prompt, images with the same number are generated using the same noise.

