# OpenReview forum: "Finetuning Text-to-Image Diffusion Models for Fairness"
_ICLR.cc/2024/Conference — ICLR 2024 oral_

### Official Review · Reviewer_KXKr · 2023-10-27

**Soundness:** 3 good
**Presentation:** 3 good
**Contribution:** 2 fair
**Rating:** 6
**Confidence:** 2

**Summary:**

The paper aims to tackle the fairness problem in text-to-image generation. They define the problem as a distribution alignment problem and use the optimal transport as the alignment loss; they also enable the user-defined target distribution as the fairness metrics. They also analyzed the straightforward end-to-end fine-tuning failure and proposed using the adjusted gradient. The experiment also shows that their method is able to reduce the bias while reserving the sematic information.

**Strengths:**

The paper is clearly written and proposes a straightforward finetuning solution to improve the fairness of the text-to-image generation model. They also propose an alignment loss based on the optimal transport and provide an analysis of the gradient of the finetuning step. Their experiments also show the effectiveness of their method.

**Weaknesses:**

This paper kind of mixing fairness and bias and uses both terms interchangeably, especially for the experiment evaluation part, they define the metric for bias by themself which reduces the credential of the evaluation. I wonder if any other metrics from other research papers have been used for evaluation. Is it possible to use well-defined fairness metrics like demographic parity/ equal opportunity, etc?

**Questions:**

1. for the equation 4, what is the u?
2. As for the alignment loss, have you tried some other metrics other than the optimal transport loss? And why did you choose the optimal transport metric?
3. For the face classifier, how accurate it is? and if the combination of face detector and face classifier is not performed well on some datasets, how does it affect the debias experiment result?

some minor comments:
1. In section 4.2, when you talk about U-net, please provide the necessary background info.

---

> ### Author Response · Authors · 2023-11-16
> **Response to Reviewer KXKr (1/2)**
>
> `Response [1/2]`
>
> Thank you for the constructive feedback, which has helped us improve the quality of our paper. We respond below to your questions and concerns:
>
> ***W1: This paper kind of mixing fairness and bias and uses both terms interchangeably, especially for the experiment evaluation part, they define the metric for bias by themself which reduces the credential of the evaluation. I wonder if any other metrics from other research papers have been used for evaluation. Is it possible to use well-defined fairness metrics like demographic parity/ equal opportunity, etc?***
>
> We apologize for the confusion arising from our uncareful use of the terms fairness and bias. They indeed have different meanings in our paper. We consider fairness as the distributional alignment with a user-defined target distribution, a flexible notion. Since our main experiments use a perfectly balanced target distribution, we define the metric bias to measure how far the generated images are from the perfectly balanced target distribution. When evaluating fairness for age, where we consider 75% young and 25% old as the target distribution, we directly report the percentage of age=old. We added clarification in the second paragraph  of section 5.1, highlighted in blue.
>
> We note that demographic parity (DP) and equal opportunity (EO) are defined for prediction problems, where every individual has a protected attribute $S$, a true label $Y$, and a prediction $\hat{Y}$ (chapter 3 of [1]). For T2I generative models, every generated image has only a protected attribute $S$. Therefore, DP and EO do not exactly apply here. In a way, the fairness problem in T2I generative models is simpler than that in prediction problems. Our objective is simply to control the generated images’ marginal distribution in terms of $S$ towards a target distribution.
>
> We understand the reviewer has concern over whether the bias metric is truthful. In response, we have added plots of gender and racial representations by each prompt in Appendix Fig. A.2, A.3, A.4, and A.5, specifically at page15 & 16. These results provide a more detailed analysis than those presented in Tab. 1.
>
>
> ***Q1: for the equation 4, what is the u?***
>
> We have significantly revised Section 4.1 to enhance the clarity of our presentation. Eq. 4 in the initial paper corresponds to Eq. 4 and 5 in the updated paper. $\\{\boldsymbol{u}_1,\cdots,\boldsymbol{u}_N\\}$ are $N$ iid samples drawn from the target distribution $\mathcal{D}$. Say we use the one-hot vectors [0,1] and [1,0] to denote male and female, and consider a balanced target distribution over these two classes. Then approximately half of $\\{\boldsymbol{u}_1,\cdots,\boldsymbol{u}_N\\}$ will be the vector [0,1] and the other half will be [1,0]. We construct $\\{\boldsymbol{u}_1,\cdots,\boldsymbol{u}_N\\}$ in this way so that it represents the target distribution and the optimal transport finds the most efficient modification of the current images towards the target distribution.
>
>
> ***Q2: As for the alignment loss, have you tried some other metrics other than the optimal transport loss? And why did you choose the optimal transport metric?***
>
> We naturally arrived at the optimal transport (OT) formulation and didn’t experiment with other losses. We start by considering the following problem: suppose we have a set of 10 images, composed of 2 female and 8 male images, and our goal is to reconfigure this set to contain an equal number of female and male images, specifically 5 of each. The key question is determining which gender class every image should be reconfigured to. Additionally, it is essential to minimise alterations to the images when striving to attain an equal distribution of 5 female and 5 male images. This is naturally the OT (or linear assignment) problem, identifying which class every image should be transported to while minimising the transport distance. Finally, the OT problem considered here is low dimensional (2 for gender, 4 for race, and 8 for gender$\times$race) and therefore efficient to solve.
>
> We sample $\\{\boldsymbol{u}_1,\cdots,\boldsymbol{u}_N\\}$ from the target distribution and compute the expectation of OT because it (1) acknowledges even a perfectly balanced distribution can still produce unbalanced finite samples, (2) takes into account the sensitivity of OT, and (3) produce a confidence measure for the target class assignment.

---

> ### Author Response · Authors · 2023-11-16
> **Response to Reviewer KXKr (2/2)**
>
> `Response [2/2]`
>
> ***Q3: For the face classifier, how accurate it is? and if the combination of face detector and face classifier is not performed well on some datasets, how does it affect the debias experiment result?***
>
> We generally require the face detector to be accurate but we did not find it a significant problem in our experiments. Mistakenly identifying other regions as faces can lead to incorrect application of the distributional alignment loss and result in erroneous edits. Therefore, we use a relatively high detection threshold for the face detector. The failure to recognize faces is less problematic. It means some faces might not be considered in our fairness objective. To reduce the frequency of this oversight, we use two different face detectors, the face_recognition [2] and the insightface [3] packages.
>
> We do not require the face classifiers to be highly accurate, such as over 95%. In our experiments, the gender classifier has 98% validation accuracy, the race classifier has 86% validation accuracy, and the age classifier has 90% validation accuracy. A few points are worth noting. Firstly, predicting certain attributes like race and age from visual cues can be inherently challenging. Secondly, the Distributional Alignment Loss (DAL) actually steers images towards those for which the classifier shows high confidence in distinguishing between male and female.  These high-confidence classifications can still provide a reliable visual guidance of gender, even when the classifier might have lower accuracy for the low-confidence classifications.
>
> ***References***
>
> [1] Solon Barocas, Moritz Hardt, and Arvind Narayanan, Fairness and Machine Learning: Limitations and Opportunities, 2023, https://fairmlbook.org
>
> [2] https://github.com/ageitgey/face_recognition
>
> [3] https://github.com/deepinsight/insightface
>
> ---
>
> Finally, we note that many other improvements have been made in the revised paper. We are happy to consider more suggestions and respond to any additional questions you may have. We would appreciate it if you could increase your score should you find our revision satisfactory.

---

> > ### Comment · Reviewer_KXKr · 2023-11-22
> >
> > Thank you for the rebuttal, it solves most of my questions. I'm not very familiar with the text-to-image area, so I'll keep my score, but I'll support the work in the discussion period.

---

> > > ### Author Response · Authors · 2023-11-22
> > > **Thank you for your feedback**
> > >
> > > Thank you for your feedback. We genuinely appreciate the time and effort you dedicated to reviewing our paper. Your support during the discussion period is highly esteemed. Should you have any further insights or inquiries, we would be pleased to continue the dialogue before the discussion period ends. Thank you once again for your thoughtful review!

---

### Official Review · Reviewer_qhuJ · 2023-10-30

**Soundness:** 4 excellent
**Presentation:** 4 excellent
**Contribution:** 4 excellent
**Rating:** 10
**Confidence:** 4

**Summary:**

The authors propose a new method for debiasing text-to-image diffusion models. While the approach is general, the authors focus on the mitigation of demographic biases here, such as gender, racial, or age-related biases in generated depictions of different occupation types or sports activities.

The method consists of a supervised fine-tuning step for different model components, which minimizes (in addition to two regularization losses that aim to preserve image quality) a loss that penalizes deviations in the demographic distribution of the generated images from a prescribed target demographic distribution.

The authors demonstrate experimentally that naive autodifferentiation-based gradient descent on this loss fails to achieve successful optimization; they provide a theoretical explanation, backed by empirical experiments, for this behavior. They then proceed to propose a simple gradient adjustment mechanism that mitigates this problem and enables successful optimization of the proposed loss function.

Finally, the authors perform extensive experiments (using Stable Diffusion as an example), an ablation study, and comparisons with previously proposed debiasing mechanisms. Throughout all experiments, the proposed method is highly successful in mitigating demographic biases for unseen prompts, especially also in regards to intersectional biases.

**Strengths:**

The manuscript addresses a topic of immediate and urgent concern.

It is very well written, appropriately references relevant prior work, and compares the proposed new method extensively to previously proposed methods.

The proposed approach is derived from first principles (distribution alignment) and intuitively appealing. The method is widely applicable - also beyond demographic debiasing - and, importantly, allows for
1) customized and explicit choices of the target data distribution, and
2) debiasing along multiple demographic dimensions at once.

The experiments are extensive, well described, and convincingly demonstrate the utility of the proposed method and its superiority over previously proposed approaches to address the same problem.

**Weaknesses:**

I could not find any major weaknesses in this manuscript. I only have a few minor questions and suggestions that I will list below.

**Questions:**

### Questions
- CLIP is not without its own issues; see e.g. Shtedritski et al., Wang et al., Wolfe et al., or Zhang and Ré. Could the authors discuss how this might affect the efficacy of their proposed method?
- How do the authors implement using different lambda values for different regions of the same image, as they describe at the end of section 4.1? ("We use a smaller weight for the non-face region ... and the smallest weight for the face region.") L_img is per image, not per pixel, no?
- I was confused about the training of the gender and race classifiers for the evaluation. Were separate classifiers trained for the training and the evaluation stage? If yes, why? Or is this maybe some kind of unintended text duplication? ("The gender and race classifiers used for the evaluation loss are trained on the CelebA and FairFace datasets. ... Evaluation. We train new gender and race classifiers using CelebA and FairFace.")
- It was unclear to me which of (soft prompt, text encoder, U-Net) were actually finetuned for all of the main experiments in section 5.1?
- Can the authors think of (and comment on) any drawbacks of applying their method at scale and for many fairness-relevant properties simultaneously? Could there be any negative consequences (e.g., in terms of image quality or biases) when e.g. generating very different images (bears, tables, groups of people, ... )? Recent observations by Qi et al. could also be interesting to discuss in this regard.

### Suggestions for improvement
- The optimal transport approach in Eq. (4) was a bit confusing to me at first because it differs from the usual setting in which the optimal transport scheme between two *distributions* is considered. By contrast, the authors consider an expectation over the optimal transport schemes between two *vectors* here. I believe everything is actually described correctly, but maybe there is a way to make this section even easier / more intuitive for readers to grasp? (Sorry, possibly not very helpful.)
- It is a little bit confusing that y denotes both the generated target labels (normal font) as well as the images generated by the frozen model (boldface).
- What does it mean that CLIP-ViT-bigG-14 and DINOv2 vit-g/14 are "more performative than the ones used in training"?
- The authors might want to consider citing Lester et al. regarding soft prompt tuning?

### References
- Lester et al., The Power of Scale for Parameter-Efficient Prompt Tuning,  https://arxiv.org/abs/2104.08691
- Qi et al., Fine-tuning Aligned Language Models Compromises Safety, Even When Users Do Not Intend To!, https://arxiv.org/abs/2310.03693
- Shtedritski et al., What does CLIP know about a red circle? Visual prompt engineering for VLMs, https://arxiv.org/abs/2304.06712
- Wang et al., FairCLIP: Social Bias Elimination based on Attribute Prototype Learning and Representation Neutralization, https://arxiv.org/abs/2210.14562
- Wolfe et al., Contrastive Language-Vision AI Models Pretrained on Web-Scraped Multimodal Data Exhibit Sexual Objectification Bias, https://dl.acm.org/doi/abs/10.1145/3593013.3594072
- Zhang and Ré, Contrastive Adapters for Foundation Model Group Robustness, https://arxiv.org/pdf/2207.07180.pdf

---

> ### Author Response · Authors · 2023-11-16
> **Response to Reviewer qhuJ (1/2)**
>
> `Response [1/2]`
>
> We are grateful and humbled by your highly positive reception of our work. We have diligently incorporated the suggestions from all reviewers into the revised paper. We respond below to your questions and concerns:
>
> ***Q1: CLIP is not without its own issues; see e.g. Shtedritski et al., Wang et al., Wolfe et al., or Zhang and Ré. Could the authors discuss how this might affect the efficacy of their proposed method?***
>
> Thank you for pointing out our oversight regarding CLIP's own biases. We added acknowledgement in Section 4.1, page 4, highlighted in blue. We did not observe significant negative impact on our debiasing method arising from CLIP and DINO’s own social impacts.  But we did observe DINO’s ViT architecture that splits an image into fixed-size patches can cause faces to appear rectangular in finetuning during our initial experiments. This observation is one of the factors that led us to use both CLIP and DINO.
>
> The inherent image-image biases of CLIP and DINO, as opposed to text-image biases, could potentially affect our debiasing methods, given that we use CLIP and DINO to minimize image-image similarities. These biases might vary from the text-image biases documented in Shtedritski et al. [1], Wang et al. [2], Wolfe et al. [3], and Zhang and Ré [4]. We hypothesise that CLIP could be more biased than DINO in our context because CLIP is trained with text supervision while DINO is trained using image self-supervision. CLIP's training process, which aligns image features with text captions, is likely to introduce social biases. Our use of both CLIP and DINO might help mitigate the impact of these biases from CLIP.
>
>
> ***Q2: How do the authors implement using different lambda values for different regions of the same image, as they describe at the end of section 4.1? ("We use a smaller weight for the non-face region ... and the smallest weight for the face region.") L_img is per image, not per pixel, no?***
>
> $\mathcal{L}\_{\textrm{img}}$ can differ between the detected face region and non-face region in the same image, and therefore is per pixel. We implement $\mathcal{L}\_{\textrm{img}}$  by adding a gradient manipulation layer right before inputting the generated image $\boldsymbol{x}^{(i)}$ to CLIP & DINO. By scaling the gradients at the level of the generated image $\boldsymbol{x}^{(i)}$, we can effectively implement a per-pixel $\mathcal{L}_{\textrm{img}}$.
>
>
> ***Q3: I was confused about the training of the gender and race classifiers for the evaluation. Were separate classifiers trained for the training and the evaluation stage? If yes, why? …***
>
> We trained separate gender and race classifiers for evaluation. We trained them for twice longer iterations, using the same architecture and training data. We do so in an effort to make the evaluation fairer. The debiasing method should not overfit to the classifier used during training and generates images that the training classifier views as fair but not the separately trained evaluation classifier. We added clarification in the revised paper.
>
>
> ***Q4: It was unclear to me which of (soft prompt, text encoder, U-Net) were actually finetuned for all of the main experiments in section 5.1?***
>
> For all of the main experiments, except explicitly stated otherwise, we finetune a LoRA adaptor with rank 50 applied on the text encoder. We added clarification in the first paragraph of Section 5.1, page 6, in the revised paper.
>
> ***Q5: Can the authors think of (and comment on) any drawbacks of applying their method at scale and for many fairness-relevant properties simultaneously? Could there be any negative consequences (e.g., in terms of image quality or biases) when e.g. generating very different images (bears, tables, groups of people, ... )? Recent observations by Qi et al. could also be interesting to discuss in this regard.***
>
> Thank you for pointing us to consider the potential negative impacts of our debiasing finetuning. In response, we added an evaluation and discussion of the debiased SD’ generations for general prompts in Appendix page 17-22. We identify the debiased SD can occasionally and to a mild extent decrease the naturalness and smoothness of the generated images. Additionally, there are limitations in the generalizability of the debiasing effect. As illustrated in Figure A.11 on page 23, for the prompt "A beautiful painting of woman by Mandy Jurgens, Trending on artstation", the debiased SD increases the representation of Asian women, but does not similarly enhance the representation of Black and Indian women. We refer the reviewer to the more detailed discussion at Appendix Section A.5, specifically at page 14 & 17, highlighted in blue.

---

> ### Author Response · Authors · 2023-11-16
> **Response to Reviewer qhuJ (2/2)**
>
> `Response [2/2]`
>
> ***Q6: The optimal transport approach in Eq. (4) was a bit confusing to me at first because it differs from the usual setting in which the optimal transport scheme between two distributions is considered. By contrast, the authors consider an expectation over the optimal transport schemes between two vectors here. I believe everything is actually described correctly, but maybe there is a way to make this section even easier / more intuitive for readers to grasp? (Sorry, possibly not very helpful.)***
>
> Your observation is correct: we consider the optimal transport (OT), or the linear assignment problem, between two sets of vectors here, followed by computing the expectation of OT w.r.t. randomness of the target set of vectors $\\{\boldsymbol{u}\_1,\cdots,\boldsymbol{u}\_N\\}$. We are particularly grateful that you pointing out our presentation of Equation 4 was not clear. We have thoroughly revised Section 4.1 to enhance its clarity. Eq. (4) in the initial paper corresponds to Eq. (4) and (5) in the revised paper.
>
>
> ***Q7: It is a little bit confusing that y denotes both the generated target labels (normal font) as well as the images generated by the frozen model (boldface).***
>
> We have changed $\boldsymbol{y}^{(i)}$ to $\boldsymbol{o}^{(i)}$ to denote the original images generated by the frozen model.
>
>
> ***Q8: What does it mean that CLIP-ViT-bigG-14 and DINOv2 vit-g/14 are "more performative than the ones used in training"?***
>
> By “more performative”, we meant the CLIP-ViT-bigG-14 used in evaluation has higher ImageNet zero-shot acc than the CLIP ViT-H/14 used in training. Similarly, the DINOv2 vit-g/14 used in evaluation also has higher ImageNet zero-shot acc than the dinov2-vit-b/14 used in training. In the revised paper, we have removed this sentence due to its lack of clarity.
>
> ***Q9: The authors might want to consider citing Lester et al. regarding soft prompt tuning?***
>
> Thank you for highlighting relevant previous research on soft prompt tuning. We have acknowledged it in the second-to-last paragraph of the introduction, highlighted in blue.
>
>
> ***References***
>
> [1] Shtedritski et al., What does CLIP know about a red circle? Visual prompt engineering for VLMs, https://arxiv.org/abs/2304.06712
>
> [2] Wang et al., FairCLIP: Social Bias Elimination based on Attribute Prototype Learning and Representation Neutralization, https://arxiv.org/abs/2210.14562
>
> [3] Wolfe et al., Contrastive Language-Vision AI Models Pretrained on Web-Scraped Multimodal Data Exhibit Sexual Objectification Bias, https://dl.acm.org/doi/abs/10.1145/3593013.3594072
>
> [4] Zhang and Ré, Contrastive Adapters for Foundation Model Group Robustness, https://arxiv.org/pdf/2207.07180.pdf
>
> ---
> Finally, we are happy to incorporate any additional feedback and provide further clarification as needed. Thank you very much for the time and effort you invested in reviewing our work.

---

> > ### Comment · Reviewer_qhuJ · 2023-11-20
> >
> > Thank you for your very thorough response and the additional experiments! I have no further comments or questions.

---

> > > ### Author Response · Authors · 2023-11-23
> > > **Thank you for your feedback**
> > >
> > > We genuinely appreciate the time and effort you dedicated to reviewing our paper. We will further polish the paper in the final revision. Thank you!

---

### Official Review · Reviewer_T8Ls · 2023-11-05

**Soundness:** 3 good
**Presentation:** 2 fair
**Contribution:** 2 fair
**Rating:** 6
**Confidence:** 2

**Summary:**

This paper studies how to finetune diffusion models with fairness as the goal. By formulating generation fairness as distribution alignments, the paper introduces a distributional alignment loss and an end-to-end fine-tuning framework. Experiments show that the fine-tuned model can generate facial images with targeted distribution of certain sensitive attributes.

**Strengths:**

1. The fairness of image generation studied in this paper is an important and practical problem.
2. The proposed fine-tuning method is sound, and the results validate the effectiveness of the method.
3. The paper has made a comprehensive analysis of which part of the model to fine-tune as well as the challenges in fine-tuning, providing insights into future fair fine-tuning work.

**Weaknesses:**

1. The paper only tested the method on single-face generation, limiting the applicability of the proposed method.
2. It is unclear whether fine-tuning with the fairness loss affects the quality and diversity of the generated images on general prompts.
3. Although experiments in the paper show better performance than baseline methods, it is unclear how expensive the fine-tuning is compared with the baseline methods.

**Questions:**

1. In Equation (4), how to obtain the expectation in practice?
2. In Table 2, on unseen prompts, how good is the proposed method compared with the baselines? Does the fine-tuning have an overfitting problem?

---

> ### Author Response · Authors · 2023-11-16
> **Response to Reviewer T8Ls (1/2)**
>
> `Response [1/2]`
>
> Thank you for the constructive feedback, which has been very helpful in enhancing the quality of our paper. We respond below to your questions and concerns:
>
> ***W1: The paper only tested the method on single-face generation, limiting the applicability of the proposed method.***
>
> We have updated our manuscript accordingly. (1) We've included an evaluation of multi-face generation in Table 3, page 7, highlighted in blue. The results show that the debiasing effect generalizes to images with multiple faces. (2) This evaluation also includes a bias metric calculated for all faces in the generated images, corroborating the original bias metric calculated using the most prominent face in each generated image. (3) Examples of generated images with multiple faces are shown in Appendix Figure A.13 & A.14, at page 29 & 30.
>
> We have also considered directly debiasing all faces in generated images but find it is more appropriate to focus on debiasing the most prominent face in each image. This is because there may be legitimate correlations among individuals within the same image, and those at the forefront should not be simply equated with those in the background. As a result, defining an ideal fair outcome for all faces in the generated images is a more intricate task. But technically, our method is capable of directly debiasing all faces in the generated images.
>
>
> ***W2: It is unclear whether fine-tuning with the fairness loss affects the quality and diversity of the generated images on general prompts.***
>
> We have incorporated an evaluation of general prompts in the revised paper, on Appendix Section A.5, page 14 & 17~22. In short, the quality of the generated images remains largely unchanged but we do identify occasional and relatively minor reduction in naturalness and smoothness of the generated images. We observed no reduction in diversity of the generated images in general. We observed increased diversity for the prompt “beautiful painting of a woman by Mandy Jurgens, trending on ArtStation”, even though the model was debiased for templated occupational prompts and never for the term "woman". We refer the reviewer to the more detailed discussion at Appendix page 14 & 17, highlighted in blue. It analyzes the quality, diversity, and fairness of images generated from general prompts.
>
>
> ***W3: Although experiments in the paper show better performance than baseline methods, it is unclear how expensive the fine-tuning is compared with the baseline methods.***
>
> Debiasing SD may require a one-time training cost or additional compute at every inference time. Our fine-tuning method generally requires more training compute than baselines. But it effectively scales to a large number of prompts and does not incur additional inference compute. In the second paragraph of the introduction (highlighted in blue), we recognized existing methods are more lightweight.
>
> In terms of specific comparisons, Ethical Intervention is a prompting method and does not incur additional training or inference compute.  Based on our experiments, DebiasVL requires around 1 GPU-hour (using an A100) training. UCE requires around 24 GPU-hours to debias 50 prompts. But we find the required compute time increases drastically when we increase the number of prompts. Concept Algebra does not require training but doubles the inference cost, which can be undesirable. Our method requires 8 GPUs * 48 hours for debiasing 1000 prompts but does not increase inference compute.
>
> We have included the training losses plot in Appendix Figure A.1, page 14. This plot indicates that although we train for 10k iterations for best performance, the finetuning achieves debiasing quite early, within 1k iterations, and can potentially be early stopped at 5k iterations. Therefore, the computational requirements for our method might not be as extensive as reported.
>
>
> ***Q1: In Equation (4), how to obtain the expectation in practice?***
>
> We have significantly revised Section 4.1 to improve its clarity. Eq. 4 in the initial paper corresponds to Eq. 4 and 5 in the revised paper. In general, the expectation is approximated by repeatedly drawing $\\{\boldsymbol{u}\_1,\cdots,\boldsymbol{u}\_N\\}$ from the target distribution $\mathcal{D}^{N}$, computing the optimal transport from $\\{\boldsymbol{p}\_1,\cdots,\boldsymbol{p}\_N\\}$ to $\\{\boldsymbol{u}\_1,\cdots,\boldsymbol{u}\_N\\}$, and computing the average of these OTs. When the number of classes $K$ as well as $N$ is small, we can enumerate all possible samples from $\mathcal{D}^{N}$, compute OT for all of them, and analytically compute the expectation by weighing different OTs according to the target distribution.

---

> ### Author Response · Authors · 2023-11-16
> **Response to Reviewer T8Ls (2/2)**
>
> `Response [2/2]`
>
> ***Q2: In Table 2, on unseen prompts, how good is the proposed method compared with the baselines? Does the fine-tuning have an overfitting problem?***
>
> We have included baselines comparisons in Table 2, page 7, highlighted in blue. We generally do not find overfitting, as demonstrated by the additional evaluation on general prompts (Appendix Section A.5, page 14 & 17~22). We did identify some negative impacts on image quality for the debiased SD, including occasional reduced naturalness and smoothness in the generated images. We discuss these in Appendix page 14 & 17, highlighted in blue.
>
> ---
>
> Finally, we note that many other improvements have been made in the revised paper. We are happy to consider more suggestions and respond to any additional questions you may have. We would appreciate it if you could increase your score should you find our revision satisfactory.

---

> ### Author Response · Authors · 2023-11-23
> **Looking forward to further feedback**
>
> Dear Reviewer T8Ls, as we are approaching the end of the author-reviewer discussion period, could you please let us know if our responses and additional experiments have adequately addressed your concerns? If there are any remaining comments, we are committed to providing prompt responses before the discussion period concludes. Thank you for your time and consideration.

---

### Author Response · Authors · 2023-11-16
**To all reviewers: we uploaded a revised version of the paper**

We sincerely thank all reviewers for their insightful and constructive feedback. In response, we have diligently incorporated the suggestions into the revised paper that we have uploaded. In the revised paper, we've highlighted the key changes in blue for easy identification, along with many other improvements. Below is a summary of the key changes, organized roughly in order of significance:
1. We added evaluation and discussion of multi-face image generation in Table 3, page 7. Examples of generated images with multiple faces are shown in Appendix Figure A.13 & A.14, at page 29 & 30.
2. We added baseline comparisons for non-templated prompts (Table 2, page 7).
3. In Appendix page 15 & 16, specifically Figs A.2~A.5, we added detailed gender and race representation plots that correspond to every entry in Table 1.
4. We show the debiased SD’s generations for general prompts in Appendix page 18-22, specifically figs.A.6~A.10. We analyze the quality, diversity, and fairness of images generated from general prompts in Appendix Section A.5, page 14 & 17.
5. We have extensively revised Section 4.1, which focuses on the design of the distributional alignment loss, to enhance its clarity.
6. We added Fig. 2 in page 6, an illustration for our finetuning scheme.
7. We have changed the term “E2E finetuning with adjusted gradient” to “adjusted direct finetuning’’ of diffusion models, adjusted DFT for short. “Direct finetuning” more closely aligns with the terminology used in existing and concurrent works that also directly finetune the diffusion model’s sampling process. “adjusted” highlights our approach of using an adjusted gradient.

---

### Meta-Review · Area_Chair_qKTm · 2023-12-08

**Metareview:**

The authors propose to unbias diffusion models by solving an additional alignment problem. Technically and empirically, the approach is very convincing and all reviewers propose acceptance.

**Justification For Why Not Higher Score:**

N/A

**Justification For Why Not Lower Score:**

The reviewers only disagree in whether this is a price worthy paper or not.

---

### Decision · Program_Chairs · 2024-01-16

Accept (oral)